# META-LEARNING WITH WARPED GRADIENT DESCENT

**Sebastian Flennerhag,**[1,2,3] **Andrei A. Rusu,**[3] **Razvan Pascanu,**[3]
**Francesco Visin,**[3] **Hujun Yin,**[1,2] **Raia Hadsell**[3]
[1]The University of Manchester, [2]The Alan Turing Institute, [3]DeepMind
`{flennerhag,andreirusu,razp,visin,raia}@google.com`
`hujun.yin@manchester.ac.uk`

## ABSTRACT

Learning an efficient update rule from data that promotes rapid learning of new tasks from the same distribution remains an open problem in meta-learning. Typically, previous works have approached this issue either by attempting to train a neural network that directly produces updates or by attempting to learn better initialisations or scaling factors for a gradient-based update rule. Both of these approaches pose challenges. On one hand, directly producing an update forgoes a useful inductive bias and can easily lead to non-converging behaviour. On the other hand, approaches that try to control a gradient-based update rule typically resort to computing gradients through the learning process to obtain their meta-gradients, leading to methods that can not scale beyond few-shot task adaptation. In this work, we propose *Warped Gradient Descent* (WarpGrad), a method that intersects these approaches to mitigate their limitations. WarpGrad meta-learns an efficiently parameterised preconditioning matrix that facilitates gradient descent across the task distribution. Preconditioning arises by interleaving non-linear layers, referred to as *warp-layers*, between the layers of a task-learner. Warp-layers are meta-learned without backpropagating through the task training process in a manner similar to methods that learn to directly produce updates. WarpGrad is computationally efficient, easy to implement, and can scale to arbitrarily large meta-learning problems. We provide a geometrical interpretation of the approach and evaluate its effectiveness in a variety of settings, including few-shot, standard supervised, continual and reinforcement learning.

## 1 INTRODUCTION

Learning (how) to learn implies inferring a learning strategy from some set of past experiences via a *meta-learner* that a *task-learner* can leverage when learning a new task. One approach is to directly parameterise an update rule via the memory of a recurrent neural network (Andrychowicz et al., 2016; Ravi & Larochelle, 2016; Li & Malik, 2016; Chen et al., 2017). Such *memory-based methods* can, in principle, represent any learning rule by virtue of being universal function approximators (Cybenko, 1989; Hornik, 1991; Schäfer & Zimmermann, 2007). They can also scale to long learning processes by using truncated backpropagation through time, but they lack an inductive bias as to what constitutes a reasonable learning rule. This renders them hard to train and brittle to generalisation as their parameter updates have no guarantees of convergence.

An alternative family of approaches defines a *gradient-based* update rule and meta-learns a shared initialisation that facilitates task adaptation across a distribution of tasks (Finn et al., 2017; Nichol et al., 2018; Flennerhag et al., 2019). Such methods are imbued with a strong inductive bias—gradient descent—but restrict knowledge transfer to the initialisation. Recent work has shown that it is beneficial to more directly control gradient descent by meta-learning an approximation of a *parameterised matrix* (Li et al., 2017; Lee & Choi, 2018; Park & Oliva, 2019) that *preconditions* gradients during task training, similarly to second-order and Natural Gradient Descent methods (Nocedal & Wright, 2006; Amari & Nagaoka, 2007). To meta-learn preconditioning, these methods backpropagate through the gradient descent process, limiting them to few-shot learning.

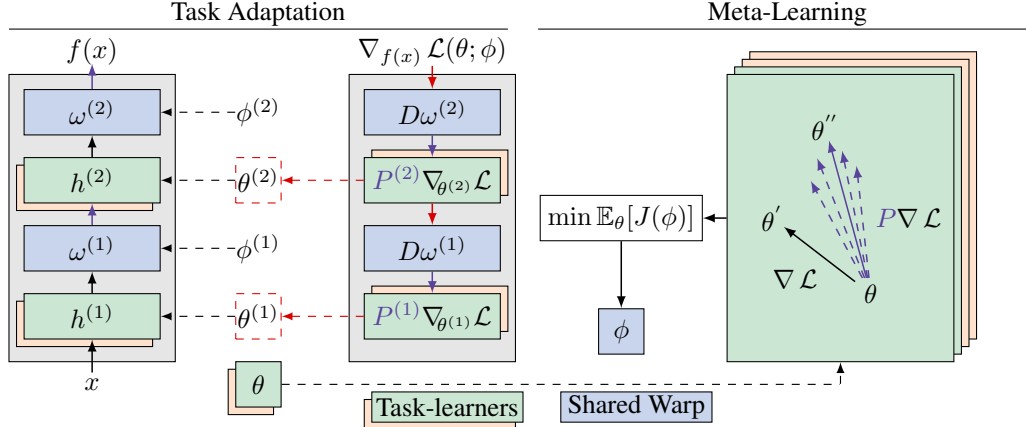

Figure 1: Schematics of WarpGrad. WarpGrad preconditioning is embedded in task-learners $f$ by interleaving warp-layers $(\omega^{(1)}, \omega^{(2)})$ between each task-learner's layers $(h^{(1)}, h^{(2)})$. WarpGrad achieve preconditioning by modulating layer activations in the forward pass and gradients in the backward pass by backpropagating through warp-layers $(D\omega)$, which implicitly preconditions gradients by some matrix $(P)$. Warp-parameters $(\phi)$ are meta-learned over the joint search space induced by task adaptation $(\mathbb{E}_\theta[J(\phi)])$ to form a geometry that facilitates task learning.

In this paper, we propose a novel framework called Warped Gradient Descent (WarpGrad)[1], that relies on the inductive bias of gradient-based meta-learners by defining an update rule that preconditions gradients, but that is meta-learned using insights from memory-based methods. In particular, we leverage that gradient preconditioning is defined point-wise in parameter space and can be seen as a recurrent operator of order 1. We use this insight to define a trajectory agnostic meta-objective over a joint parameter search space where knowledge transfer is encoded in gradient preconditioning.

To achieve a scalable and flexible form of preconditioning, we take inspiration from works that embed preconditioning in task-learners (Desjardins et al., 2015; Lee & Choi, 2018), but we relax the assumption that task-learners are feed-forward and replace their linear projection with a generic neural network $\omega$, referred to as a *warp layer*. By introducing non-linearity, preconditioning is rendered data-dependent. This allows WarpGrad to model preconditioning beyond the block-diagonal structure of prior works and enables it to meta-learn over arbitrary adaptation processes.

We empirically validate WarpGrad and show it surpasses baseline gradient-based meta-learners on standard few-shot learning tasks (*mini*ImageNet, *tiered*ImageNet; Vinyals et al., 2016; Ravi & Larochelle, 2016; Ren et al., 2018), while scaling beyond few-shot learning to standard supervised settings on the "multi"-shot Omniglot benchmark (Flennerhag et al., 2019) and a multi-shot version of *tiered*ImageNet. We further find that WarpGrad outperforms competing methods in a reinforcement learning (RL) setting where previous gradient-based meta-learners fail (maze navigation with recurrent neural networks (Miconi et al., 2019)) and can be used to meta-learn an optimiser that prevents catastrophic forgetting in a continual learning setting.

## 2 WARPED GRADIENT DESCENT

### 2.1 GRADIENT-BASED META-LEARNING

WarpGrad belongs to the family of optimisation-based meta-learners that parameterise an update rule $\theta \leftarrow U(\theta; \xi)$ with some meta-parameters $\xi$. Specifically, gradient-based meta-learners define an update rule by relying on the gradient descent, $U(\theta; \xi) := \theta - \alpha \nabla \mathcal{L}(\theta)$ for some objective $\mathcal{L}$ and learning rate $\alpha$. A task is defined by a training set $\mathcal{D}_{\text{train}}^\tau$ and a test set $\mathcal{D}_{\text{test}}^\tau$, which defines learning objectives $\mathcal{L}_{\mathcal{D}^\tau}(\theta) := \mathbb{E}_{(x,y) \sim \mathcal{D}^\tau}[\ell(f(x, \theta), y)]$ over the task-learner $f$ for some loss $\ell$. MAML (Finn

---

[1]Open-source implementation available at https://github.com/flennerhag/warpgrad.

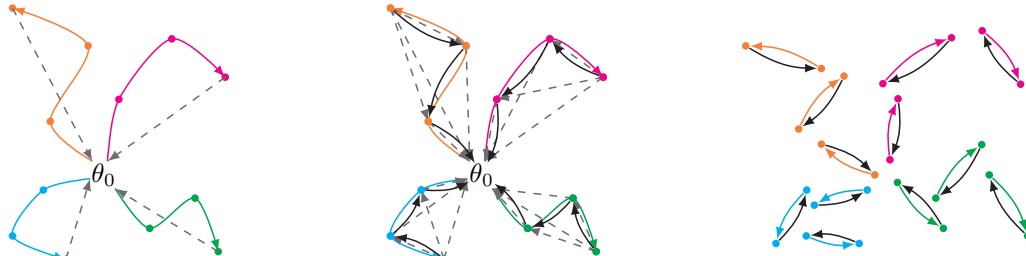

Figure 2: Gradient-based meta-learning. Colours denote different tasks ($\tau$), dashed lines denote backpropagation through the adaptation process, and solid black lines denote optimiser parameter ($\phi$) gradients w.r.t. one step of task parameter ($\theta$) adaptation. *Left:* A meta-learned initialisation compresses trajectory information into a single initial point ($\theta_0$). *Middle:* MAML-based optimisers interact with adaptation trajectories at every step and backpropagate through each interaction. *Right:* WarpGrad is trajectory agnostic. Task adaptation defines an empirical distribution $p(\tau, \theta)$ over which WarpGrad learns a geometry for adaptation by optimising for steepest descent directions.

et al., 2017) meta-learns a shared initialisation $\theta_0$ by backpropagating through $K$ steps of gradient descent across a given task distribution $p(\tau)$,

$$C^{\text{MAML}}(\xi) \coloneqq \sum_{\tau \sim p(\tau)} \mathcal{L}_{\mathcal{D}_{\text{test}}^{\tau}} \left( \theta_0 - \alpha \sum_{k=0}^{K-1} U_{\mathcal{D}_{\text{train}}^{\tau}}(\theta_k^{\tau}; \xi) \right). \tag{1}$$

Subsequent works on gradient-based meta-learning primarily differ in the parameterisation of $U$. Meta-SGD (MSGD; Li & Malik, 2016) learns a vector of learning rates, whereas Meta-Curvature (MC; Park & Oliva, 2019) defines a block-diagonal preconditioning matrix $B$, and T-Nets (Lee & Choi, 2018) embed block-diagonal preconditioning in feed-forward learners via linear projections,

$$U(\theta_k; \theta_0) \coloneqq \theta_k - \alpha \nabla \mathcal{L}(\theta_k) \qquad\qquad \text{MAML} \tag{2}$$
$$U(\theta_k; \theta_0, \phi) \coloneqq \theta_k - \alpha \operatorname{diag}(\phi) \nabla \mathcal{L}(\theta_k) \qquad\qquad \text{MSGD} \tag{3}$$
$$U(\theta_k; \theta_0, \phi) \coloneqq \theta_k - \alpha B(\theta_k; \phi) \nabla \mathcal{L}(\theta_k) \qquad\qquad \text{MC} \tag{4}$$
$$U(\theta_k; \theta_0, \phi) \coloneqq \theta_k - \alpha \nabla \mathcal{L}(\theta_k; \phi) \qquad\qquad \text{T-Nets.} \tag{5}$$

These methods optimise for meta-parameters $\xi = \{\theta_0, \phi\}$ by backpropagating through the gradient descent process (Eq. 1). This trajectory dependence limits them to few-shot learning as they become (1) computationally expensive, (2) susceptible to exploding/vanishing gradients, and (3) susceptible to a credit assignment problem (Wu et al., 2018; Antoniou et al., 2019; Liu et al., 2019).

Our goal is to develop a meta-learner that overcomes all three limitations. To do so, we depart from the paradigm of backpropagating to the initialisation and exploit the fact that learning to precondition gradients can be seen as a Markov Process of order 1 that depends on the state but not the trajectory (Li et al., 2017). To develop this notion, we first establish a general-purpose form of preconditioning (Section 2.2). Based on this, we obtain a canonical meta-objective from a geometrical point of view (Section 2.3), from which we derive a trajectory-agnostic meta-objective (Section 2.4).

## 2.2 General-Purpose Preconditioning

A preconditioned gradient descent rule, $U(\theta; \phi) \coloneqq \theta - \alpha P(\theta; \phi) \nabla \mathcal{L}(\theta)$, defines a geometry via $P$. To disentangle the expressive capacity of this geometry from the expressive capacity of the task-learner $f$, we take inspiration from T-Nets that embed linear projections $T$ in feed-forward layers, $h = \sigma(TWx + b)$. This in itself is not sufficient to achieve disentanglement since the parameterisation of $T$ is directly linked to that of $W$, but it can be achieved under non-linear preconditioning.

To this end, we relax the assumption that the task-learner is feed-forward and consider an arbitrary neural network, $f = h^{(L)} \circ \cdots \circ h^{(1)}$. We insert warp-layers that are universal function approximators parameterised by neural networks into the task-learner without restricting their form or how they

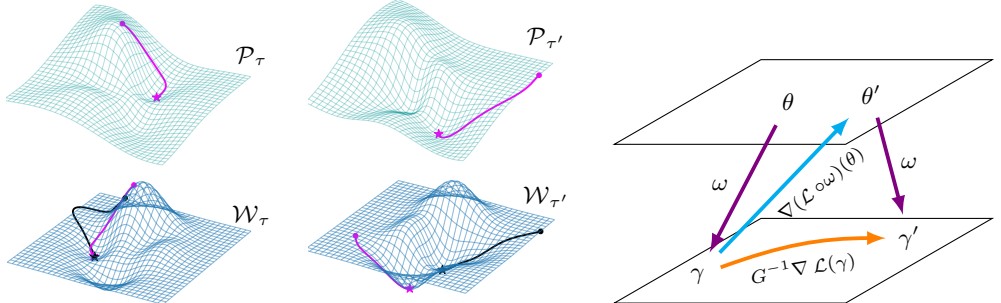

Figure 3: *Left:* synthetic experiment illustrating how WarpGrad warps gradients (see Appendix D for full details). Each task $f \sim p(f)$ defines a distinct loss surface ($\mathcal{W}$, bottom row). Gradient descent (black) on these surfaces struggles to find a minimum. WarpGrad meta-learns a *warp* $\omega$ to produce better update directions (magenta; Section 2.4). In doing so, WarpGrad learns a meta-geometry $\mathcal{P}$ where standard gradient descent is well behaved (top row). *Right:* gradient descent in $\mathcal{P}$ is equivalent to first-order Riemannian descent in $\mathcal{W}$ under a meta-learned Riemann metric (Section 2.3).

interact with $f$. In the simplest case, we interleave warp-layers between layers of the task-learner to obtain $\hat{f} = \omega^{(L)} \circ h^{(L)} \circ \cdots \circ \omega^{(1)} \circ h^{(1)}$, but other forms of interaction can be beneficial (see Appendix A for practical guidelines). Backpropagation automatically induces gradient preconditioning, as in T-Nets, but in our case via the Jacobians of the warp-layers:

$$\frac{\partial \mathcal{L}}{\partial \theta^{(i)}} = \mathbb{E}\left[\nabla \ell^T \left(\prod_{j=0}^{L-(i+1)} D_x \omega^{(L-j)} D_x h^{(L-j)}\right) D_x \omega^{(i)} D_\theta h^{(i)}\right], \quad (6)$$

where $D_x$ and $D_\theta$ denote the Jacobian with respect to input and parameters, respectively. In the special case where $f$ is feed-forward and each $\omega$ a linear projection, we obtain an instance of WarpGrad that is akin to T-Nets since preconditioning is given by $D_x \omega = T$. Conversely, by making warp-layers non-linear, we can induce interdependence between warp-layers, allowing WarpGrad to model preconditioning beyond the block-diagonal structure imposed by prior works. Further, this enables a form of task-conditioning by making Jacobians of warp-layers data dependent. As we have made no assumptions on the form of the task-learner or warp-layers, WarpGrad methods can act on any neural network through any form of warping, including recurrence. We show that increasing the capacity of the meta-learner by defining warp-layers as Residual Networks (He et al., 2017) improves performance on classification tasks (Section 4.1). We also introduce recurrent warp-layers for agents in a gradient-based meta-learner that is the first, to the best of our knowledge, to outperform memory-based meta-learners on a maze navigation task that requires memory (Section 4.3).

Warp-layers imbue WarpGrad with three powerful properties. First, due to preconditioned gradients, WarpGrad inherits gradient descent properties, importantly guarantees of convergence. Second, warp-layers form a distributed representation of preconditioning that disentangles the expressiveness of the geometry it encodes from the expressive capacity of the task-learner. Third, warp-layers are meta-learned across tasks and trajectories and can therefore capture properties of the task-distribution beyond local information. Figure 3 illustrates these properties in a synthetic scenario, where we construct a family of tasks $f : \mathbb{R}^2 \to \mathbb{R}$ (see Appendix D for details) and meta-learn across the task distribution. WarpGrad learns to produce warped loss surfaces (illustrated on two tasks $\tau$ and $\tau'$) that are smoother and more well-behaved than their respective native loss-surfaces.

## 2.3 THE GEOMETRY OF WARPED GRADIENT DESCENT

If the preconditioning matrix $P$ is invertible, it defines a valid Riemann metric (Amari, 1998) and therefore enjoys similar convergence guarantees to gradient descent. Thus, if warp-layers represent

a valid (meta-learned) Riemann metric, WarpGrad is well-behaved. For T-Nets, it is sufficient to require $T$ to be full rank, since $T$ explicitly defines $P$ as a block-diagonal matrix with block entries $TT^T$. In contrast, non-linearity in warp-layers precludes such an explicit identification.

Instead, we must consider the geometry that warp-layers represent. For this, we need a metric tensor, $G$, which is a positive-definite, smoothly varying matrix that measures curvature on a manifold $\mathcal{W}$. The metric tensor defines the steepest direction of descent by $-G^{-1}\nabla\mathcal{L}$ (Lee, 2003), hence our goal is to establish that warp-layers approximate some $G^{-1}$. Let $\Omega$ represent the effect of warp-layers by a reparameterisation $h^{(i)}(x;\Omega(\theta;\phi)^{(i)}) = \omega^{(i)}(h^{(i)}(x;\theta^{(i)});\phi) \ \forall x, i$ that maps from a space $\mathcal{P}$ onto the manifold $\mathcal{W}$ with $\gamma = \Omega(\theta;\phi)$. We induce a metric $G$ on $\mathcal{W}$ by push-forward (Figure 2):

$$\Delta\theta \coloneqq \nabla\left(\mathcal{L}\circ\Omega\right)(\theta;\phi) = [D_x\Omega(\theta;\phi)]^T\nabla\mathcal{L}(\gamma) \qquad \mathcal{P}\text{-space} \qquad (7)$$

$$\Delta\gamma \coloneqq D_x\Omega(\theta;\phi)\,\Delta\theta = G(\gamma;\phi)^{-1}\nabla\mathcal{L}(\gamma) \qquad \mathcal{W}\text{-space}, \qquad (8)$$

where $G^{-1} \coloneqq [D_x\Omega][D_x\Omega]^T$. Provided $\Omega$ is not degenerate ($G$ is non-singular), $G^{-1}$ is positive-definite, hence a valid Riemann metric. While this is the metric induced on $\mathcal{W}$ by warp-layers, it is not the metric used to precondition gradients since we take gradient steps in $\mathcal{P}$ which introduces an error term (Figure 2). We can bound the error by first-order Taylor series expansion to establish first-order equivalence between the WarpGrad update in $\mathcal{P}$ (Eq. 7) and the ideal update in $\mathcal{W}$ (Eq. 8),

$$(\mathcal{L}\circ\Omega)(\theta - \alpha\Delta\theta) = \mathcal{L}(\gamma - \alpha\Delta\gamma) + \mathcal{O}(\alpha^2). \qquad (9)$$

Consequently, gradient descent under warp-layers (in $\mathcal{P}$-space) is first-order equivalent to warping the native loss surface under a metric $G$ to facilitate task adaptation. Warp parameters $\phi$ control the geometry induced by warping, and therefore *what* task-learners converge to. By meta-learning $\phi$ we can accumulate information that is conducive to task adaptation but that may not be available during that process. This suggests that an ideal geometry (in $\mathcal{W}$-space) should yield preconditioning that points in the direction of steepest descent, accounting for global information across tasks,

$$\min_\phi \ \mathbb{E}_{\mathcal{L},\gamma\sim p(\mathcal{L},\gamma)}\left[\mathcal{L}\left(\gamma - \alpha\,G(\gamma;\phi)^{-1}\nabla\mathcal{L}(\gamma)\right)\right]. \qquad (10)$$

In contrast to MAML-based approaches (Eq. 1), this objective avoids backpropagation through learning processes. Instead, it defines task learning abstractly by introducing a joint distribution over objectives and parameterisations, opening up for general-purpose meta-learning at scale.

## 2.4   META-LEARNING WARP PARAMETERS

The canonical objective in Eq. 10 describes a meta-objective for learning a geometry on first principles that we can render into a trajectory-agnostic update rule for warp-layers. To do so, we define a task $\tau = (h^\tau, \mathcal{L}^\tau_{\text{meta}}, \mathcal{L}^\tau_{\text{task}})$ by a task-learner $\hat{f}$ that is embedded with a shared WarpGrad optimiser, a meta-training objective $\mathcal{L}^\tau_{\text{meta}}$, and a task adaptation objective $\mathcal{L}^\tau_{\text{task}}$. We use $\mathcal{L}^\tau_{\text{task}}$ to adapt task parameters $\theta$ and $\mathcal{L}^\tau_{\text{meta}}$ to adapt warp parameters $\phi$. Note that we allow meta and task objectives to differ in arbitrary ways, but both are expectations over some data, as above. In the simplest case, they differ in terms of validation versus training data, but they may differ in terms of learning paradigm as well, as we demonstrate in continual learning experiment (Section 4.3).

To obtain our meta-objective, we recast the canonical objective (Eq. 10) in terms of $\theta$ using first-order equivalence of gradient steps (Eq. 9). Next, we factorise $p(\tau, \theta)$ into $p(\theta \mid \tau)p(\tau)$. Since $p(\tau)$ is given, it remains to consider a sampling strategy for $p(\theta \mid \tau)$. For meta-learning of warp-layers, we assume this distribution is given. We later show how to incorporate meta-learning of a prior $p(\theta_0 \mid \tau)$. While any sampling strategy is valid, in this paper we exploit that task learning under stochastic gradient descent can be seen as sampling from an empirical prior $p(\theta \mid \tau)$ (Grant et al., 2018); in particular, each iterate $\theta^\tau_k$ can be seen as a sample from $p(\theta^\tau_k \mid \theta^\tau_{k-1}, \phi)$. Thus, $K$-steps of gradient descent forms a Monte-Carlo chain $\theta^\tau_0, \ldots, \theta^\tau_K$ and sampling such chains define an empirical distribution $p(\theta \mid \tau)$ around some prior $p(\theta_0 \mid \tau)$, which we will discuss in Section 2.5. The joint distribution $p(\tau, \theta)$ defines a joint search space across tasks. Meta-learning therefore learns

a geometry over this space with the steepest expected direction of descent. This direction is however not with respect to the objective that produced the gradient, $\mathcal{L}_{\text{task}}^\tau$, but with respect to $\mathcal{L}_{\text{meta}}^\tau$,

$$L(\phi) := \sum_{\tau \sim p(\tau)} \sum_{\theta^\tau \sim p(\theta|\tau)} \mathcal{L}_{\text{meta}}^\tau \left( \theta^\tau - \alpha \nabla \mathcal{L}_{\text{task}}^\tau (\theta^\tau; \phi); \phi \right). \tag{11}$$

Decoupling the task gradient operator $\nabla \mathcal{L}_{\text{task}}^\tau$ from the geometry learned by $\mathcal{L}_{\text{meta}}^\tau$ lets us infuse global knowledge in warp-layers, a promising avenue for future research (Metz et al., 2019; Mendonca et al., 2019). For example, in Section 4.3, we meta-learn an update-rule that mitigates catastrophic forgetting by defining $\mathcal{L}_{\text{meta}}^\tau$ over current and previous tasks. In contrast to other gradient-based meta-learners, the WarpGrad meta-objective is an expectation over gradient update steps sampled from the search space induced by task adaptation (for example, $K$ steps of stochastic gradient descent; Figure 2). It is therefore trajectory agnostic and hence compatible with arbitrary task learning processes. Because the meta-gradient is independent of the number of task gradient steps, it avoids vanishing/exploding gradients and the credit assignment problem by design. It does rely on second-order gradients, a requirement we can relax by detaching task parameter gradients ($\nabla \mathcal{L}_{\text{task}}^\tau$) in Eq. 11,

$$\hat{L}(\phi) := \sum_{\tau \sim p(\tau)} \sum_{\theta^\tau \sim p(\theta|\tau)} \mathcal{L}_{\text{meta}}^\tau \left( \text{sg} \left[ \theta^\tau - \alpha \nabla \mathcal{L}_{\text{task}}^\tau (\theta^\tau; \phi) \right]; \phi \right), \tag{12}$$

where sg is the stop-gradient operator. In contrast to the first-order approximation of MAML (Finn et al., 2017), which ignores the entire trajectory except for the final gradient, this approximation retains all gradient terms and only discards local second-order effects, which are typically dominated by first-order effect in long parameter trajectories (Flennerhag et al., 2019). Empirically, we find that our approximation only incurs a minor loss of performance in an ablation study on Omniglot (Appendix F). Interestingly, this approximation is a form of multitask learning with respect to $\phi$ (Li & Hoiem, 2016; Bilen & Vedaldi, 2017; Rebuffi et al., 2017) that marginalises over task parameters $\theta^\tau$.

| **Algorithm 1** WarpGrad: online meta-training | **Algorithm 2** WarpGrad: offline meta-training |
|---|---|
| **Require:** $p(\tau)$: distribution over tasks | **Require:** $p(\tau)$: distribution over tasks |
| **Require:** $\alpha, \beta, \lambda$: hyper-parameters | **Require:** $\alpha, \beta, \lambda, \eta$: hyper-parameters |
| 1: initialise $\phi$ and $p(\theta_0 \mid \tau)$ | 1: initialise $\phi, p(\theta_0 \mid \tau)$ |
| 2: **while** not done **do** | 2: **while** not done **do** |
| 3:    sample mini-batch of tasks $\mathcal{T}$ from $p(\tau)$ | 3:    initialise buffer $\mathcal{B} = \{\}$ |
| 4:    $g_\phi, g_{\theta_0} \leftarrow 0$ | 4:    sample mini-batch of tasks $\mathcal{T}$ from $p(\tau)$ |
| 5:    **for all** $\tau \in \mathcal{T}$ **do** | 5:    **for all** $\tau \in \mathcal{T}$ **do** |
| 6:       $\theta_0^\tau \sim p(\theta_0 \mid \tau)$ | 6:       $\theta_0^\tau \sim p(\theta_0 \mid \tau)$ |
| 7:       **for all** $k$ in $0, \ldots, K_\tau - 1$ **do** | 7:       $\mathcal{B}[\tau] = [\theta_0^\tau]$ |
| 8:          $\theta_{k+1}^\tau \leftarrow \theta_k^\tau - \alpha \nabla \mathcal{L}_{\text{task}}^\tau (\theta_k^\tau; \phi)$ | 8:       **for all** $k$ in $0, \ldots, K_\tau - 1$ **do** |
| 9:          $g_\phi \leftarrow g_\phi + \nabla L(\phi; \theta_k^\tau)$ | 9:          $\theta_{k+1}^\tau \leftarrow \theta_k^\tau - \alpha \nabla \mathcal{L}_{\text{task}}^\tau (\theta_k^\tau; \phi)$ |
| 10:         $g_{\theta_0} \leftarrow g_{\theta_0} + \nabla C(\theta_0; \theta_{0:k}^\tau)$ | 10:         $\mathcal{B}[\tau].\text{append}(\theta_{k+1}^\tau)$ |
| 11:       **end for** | 11:       **end for** |
| 12:    **end for** | 12:    **end for** |
| 13:    $\phi \leftarrow \phi - \beta g_\phi$ | 13:    $i, g_\phi, g_{\theta_0} \leftarrow 0$ |
| 14:    $\theta_0 \leftarrow \theta_0 - \lambda \beta g_{\theta_0}$ | 14:    **for all** $(\tau, k) \in \mathcal{B}$ **do** |
| 15: **end while** | 15:       $g_\phi \leftarrow g_\phi + \nabla L(\phi; \theta_k^\tau)$ |
| | 16:       $g_{\theta_0} \leftarrow g_{\theta_0} + \nabla C(\theta_0^\tau; \theta_{0:k}^\tau)$ |
| | 17:       $i \leftarrow i + 1$ |
| | 18:       **if** $i = \eta$ **then** |
| | 19:          $\phi \leftarrow \phi - \beta g_\phi$ |
| | 20:          $\theta_0 \leftarrow \theta_0 - \lambda \beta g_{\theta_0}$ |
| | 21:          $i, g_\phi, g_{\theta_0} \leftarrow 0$ |
| | 22:       **end if** |
| | 23:    **end for** |
| | 24: **end while** |

### 2.5 Integration with Learned Initialisations

WarpGrad is a method for learning warp layer parameters $\phi$ over a joint search space defined by $p(\tau, \theta)$. Because WarpGrad takes this distribution as given, we can integrate WarpGrad with methods that define or learn some form of "prior" $p(\theta_0 \mid \tau)$ over $\theta_0^\tau$. For instance, (a) *Multi-task solution*: in online learning, we can alternate between updating a multi-task solution and tuning warp parameters. We use this approach in our Reinforcement Learning experiment (Section 4.3); (b) *Meta-learned point-estimate*: when task adaptation occurs in batch mode, we can meta-learn a shared initialisation $\theta_0$. Our few-shot and supervised learning experiments take this approach (Section 4.1); (c) *Meta-learned prior*: WarpGrad can be combined with Bayesian methods that define a full prior (Rusu et al., 2019; Oreshkin et al., 2018; Lacoste et al., 2018; Kim et al., 2018). We incorporate such methods by some objective $C$ (potentially vacuous) over $\theta_0$ that we optimise jointly with WarpGrad,

$$J(\phi, \theta_0) := L(\phi) + \lambda C(\theta_0),\qquad(13)$$

where $L$ can be substituted for by $\hat{L}$ and $\lambda \in [0, \infty)$ is a hyper-parameter. We train the WarpGrad optimiser via stochastic gradient descent and solve Eq. 13 by alternating between sampling task parameters from $p(\tau, \theta)$ given the current parameter values for $\phi$ and taking meta-gradient steps over these samples to update $\phi$. As such, our method can also be seen as a generalised form of gradient descent in the form of Mirror Descent with a meta-learned dual space (Desjardins et al., 2015; Beck & Teboulle, 2003). The details of the sampling procedure may vary depending on the specifics of the tasks (static, sequential), the design of the task-learner (feed-forward, recurrent), and the learning objective (supervised, self-supervised, reinforcement learning). In Algorithm 1 we illustrate a simple online algorithm with constant memory and linear complexity in $K$, assuming the same holds for $C$. A drawback of this approach is that it is relatively data inefficient; in Appendix B we detail a more complex offline training algorithm that stores task parameters in a replay buffer for mini-batched training of $\phi$. The gains of the offline variant can be dramatic: in our Omniglot experiment (Section 4.1), offline meta-training allows us to update warp parameters 2000 times with each meta-batch, improving final test accuracy from $76.3\%$ to $84.3\%$ (Appendix F).

## 3 Related Work

Learning to learn, or meta-learning, has previously been explored in a variety of settings. Early work focused on evolutionary approaches (Schmidhuber, 1987; Bengio et al., 1991; Thrun & Pratt, 1998). Hochreiter et al. (2001) introduced gradient descent methods to meta-learning, specifically for recurrent meta-learning algorithms, extended to RL by Wang et al. (2016) and Duan et al. (2016). A similar approach was taken by Andrychowicz et al. (2016) and Ravi & Larochelle (2016) to meta-learn a parameterised update rule in the form of a Recurrent Neural Network (RNN).

A related idea is to separate parameters into "slow" and "fast" weights, where the former captures meta-information and the latter encapsulates rapid adaptation (Hinton & Plaut, 1987; Schmidhuber, 1992; Ba et al., 2016). This can be implemented by embedding a neural network that dynamically adapts the parameters of a main architecture (Ha et al., 2016). WarpGrad can be seen as learning slow warp-parameters that precondition adaptation of fast weights. Recent meta-learning focuses almost exclusively on few-shot learning, where tasks are characterised by severe data scarcity. In this setting, tasks must be sufficiently similar that a new task can be learned from a single or handful of examples (Lake et al., 2015; Vinyals et al., 2016; Snell et al., 2017; Ren et al., 2018).

Several meta-learners have been proposed that directly predict the parameters of the task-learner (Bertinetto et al., 2016; Munkhdalai et al., 2018; Gidaris & Komodakis, 2018; Qiao et al., 2018). To scale, such methods typically pretrain a feature extractor and predict a small subset of the parameters. Closely related to our work are gradient-based few-shot learning methods that extend MAML by sharing some subset of parameters between task-learners that is fixed during task training but meta-learner across tasks, which may reduce overfitting (Mishra et al., 2018; Lee & Choi, 2018; Munkhdalai et al., 2018) or induce more robust convergence (Zintgraf et al., 2019). It can also be used to model latent variables for concept or task inference, which implicitly induce gradient modulation (Zhou et al., 2018; Oreshkin et al., 2018; Rusu et al., 2019; Lee et al., 2019). Our work is also related to gradient-based meta-learning of a shared initialisation that scales beyond few-shot learning (Nichol et al., 2018; Flennerhag et al., 2019).

Meta-learned preconditioning is closely related to parallel work on second-order optimisation methods for high dimensional non-convex loss surfaces (Nocedal & Wright, 2006; Saxe et al., 2013; Kingma & Ba, 2015; Arora et al., 2018). In this setting, second-order optimisers typically struggle to improve upon first-order baselines (Sutskever et al., 2013). As second-order curvature is typically intractable to compute, such methods resort to low-rank approximations (Nocedal & Wright, 2006; Martens, 2010; Martens & Grosse, 2015) and suffer from instability (Byrd et al., 2016). In particular, Natural Gradient Descent (Amari, 1998) is a method that uses the Fisher Information Matrix as curvature metric (Amari & Nagaoka, 2007). Several proposed methods for amortising the cost of estimating the metric (Pascanu & Bengio, 2014; Martens & Grosse, 2015; Desjardins et al., 2015). As noted by Desjardins et al. (2015), expressing preconditioning through interleaved projections can be seen as a form of Mirror Descent (Beck & Teboulle, 2003). WarpGrad offers a new perspective on gradient preconditioning by introducing a generic form of model-embedded preconditioning that exploits global information beyond the task at hand.

# 4 EXPERIMENTS

We evaluate WarpGrad in a set of experiments designed to answer three questions: (1) do WarpGrad methods retain the inductive bias of MAML-based few-shot learners? (2) Can WarpGrad methods scale to problems beyond the reach of such methods? (3) Can WarpGrad generalise to complex meta-learning problems?

## 4.1 FEW-SHOT LEARNING

For few-shot learning, we test whether WarpGrad retains the inductive bias of gradient-based meta-learners while avoiding backpropagation through the gradient descent process. To isolate the effect of the WarpGrad objective, we use *linear* warp-layers that we train using online meta-training (Algorithm 1) to make WarpGrad as close to T-Nets as possible. For a fair comparison, we meta-learn the initialisation using MAML (Warp-MAML) with $J(\theta_0, \phi) := L(\phi) + \lambda C^{\text{MAML}}(\theta_0)$. We evaluate the importance of meta-learning the initialisation in Appendix G and find that WarpGrad achieves similar performance under random task parameter initialisation.

Table 1: Mean test accuracy after task adaptation on held out evaluation tasks. [†]Multi-headed. [‡]No meta-training; see Appendix E and Appendix H.

| | *mini*ImageNet | |
| --- | --- | --- |
| | 5-way 1-shot | 5-way 5-shot |
| Reptile | $50.0 \pm 0.3$ | $66.0 \pm 0.6$ |
| Meta-SGD | $50.5 \pm 1.9$ | $64.0 \pm 0.9$ |
| (M)T-Net | $51.7 \pm 1.8$ | – |
| CAVIA (512) | $51.8 \pm 0.7$ | $65.9 \pm 0.6$ |
| MAML | $48.7 \pm 1.8$ | $63.2 \pm 0.9$ |
| **Warp-MAML** | **$52.3 \pm 0.8$** | **$68.4 \pm 0.6$** |

| | *tiered*ImageNet | |
| --- | --- | --- |
| | 5-way 1-shot | 5-way 5-shot |
| MAML | $51.7 \pm 1.8$ | $70.3 \pm 1.8$ |
| **Warp-MAML** | **$57.2 \pm 0.9$** | **$74.1 \pm 0.7$** |

| | *tiered*ImageNet | Omniglot |
| --- | --- | --- |
| | 10-way 640-shot | 20-way 100-shot |
| SGD[‡] | $58.1 \pm 1.5$ | $51.0$ |
| KFAC[‡] | – | $56.0$ |
| Finetuning[†] | – | $76.4 \pm 2.2$ |
| Reptile | $76.52 \pm 2.1$ | $70.8 \pm 1.9$ |
| Leap | $73.9 \pm 2.2$ | $75.5 \pm 2.6$ |
| **Warp-Leap** | **$80.4 \pm 1.6$** | **$83.6 \pm 1.9$** |

All task-learners use a convolutional architecture that stacks 4 blocks made up of a $3 \times 3$ convolution, max-pooling, batch-norm, and ReLU activation. We define Warp-MAML by inserting warp-layers in the form of $3 \times 3$ convolutions after each block in the baseline task-learner. All baselines are tuned with identical and independent hyper-parameter searches (including filter sizes – full experimental settings in Appendix H), and we report best results from our experiments or the literature. Warp-MAML outperforms all baselines (Table 1), improving 1- and 5-shot accuracy by 3.6 and 5.5 percentage points on *mini*ImageNet (Vinyals et al., 2016; Ravi & Larochelle, 2016) and by 5.2 and 3.8 percentage points on *tiered*ImageNet (Ren et al., 2018), which indicates that WarpGrad retains the inductive bias of MAML-based meta-learners.

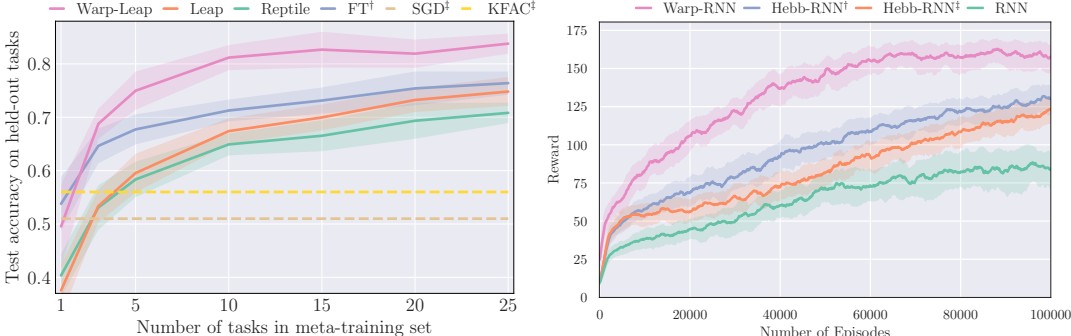

Figure 4: *Left:* Omniglot test accuracies on held-out tasks after meta-training on a varying number of tasks. Shading represents standard deviation across 10 independent runs. We compare Warp-Leap, Leap, and Reptile, multi-headed finetuning, as well as SGD and KFAC which used random initialisation but with 4x larger batch size and 10x larger learning rate. *Right:* On a RL maze navigation task, mean cumulative return is shown. Shading represents inter-quartile ranges across 10 independent runs.[†]Simple modulation and [‡]retroactive modulation are used (Miconi et al., 2019).

## 4.2 MULTI-SHOT LEARNING

Next, we evaluate whether WarpGrad can scale beyond few-shot adaptation on similar supervised problems. We propose a new protocol for *tiered*ImageNet that increases the number of adaptation steps to 640 and use 6 convolutional blocks in task-learners, which are otherwise defined as above. Since MAML-based approaches cannot backpropagate through 640 adaptation steps for models of this size, we evaluate WarpGrad against two gradient-based meta-learners that meta-learn an initialisation without such backpropagation, Reptile (Nichol et al., 2018) and Leap (Flennerhag et al., 2019), and we define a Warp-Leap meta-learner by $J(\theta_0, \phi) := L(\phi) + \lambda C^{\text{Leap}}(\theta_0)$. Leap is an attractive complement as it minimises the expected gradient descent trajectory length across tasks. Under WarpGrad, this becomes a joint search for a geometry in which task adaptation defines geodesics (shortest paths, see Appendix C for details). While Reptile outperforms Leap by 2.6 percentage points on this benchmark, Warp-Leap surpasses both, with a margin of 3.88 to Reptile (Table 1).

We further evaluate Warp-Leap on the multi-shot Omniglot (Lake et al., 2011) protocol proposed by Flennerhag et al. (2019), where each of the 50 alphabets is a 20-way classification task. Task adaptation involves 100 gradient steps on random samples that are preprocessed by random affine transformations. We report results for Warp-Leap under offline meta-training (Algorithm 2), which updates warp parameters 2000 times per meta step (see Appendix E for experimental details). Warp-Leap enjoys similar performance on this task as well, improving over Leap and Reptile by 8.1 and 12.8 points respectively (Table 1). We also perform an extensive ablation study varying the number of tasks in the meta-training set. Except for the case of a single task, Warp-Leap substantially outperforms all baselines (Figure 4), achieving a higher rate of convergence and reducing the final test error from ~30% to ~15% . Non-linear warps, which go beyond block-diagonal preconditioning, reach ~11% test error (refer to Appendix F and Table 2 for the full results). Finally, we find that WarpGrad methods behave distinctly different from Natural Gradient Descent methods in an ablation study (Appendix G). It reduces final test error from ~42% to ~19%, controlling for initialisation, while its preconditioning matrices differ from what the literature suggests (Desjardins et al., 2015).

## 4.3 COMPLEX META-LEARNING: REINFORCEMENT AND CONTINUAL LEARNING

**(c.1) Reinforcement Learning** To illustrate how WarpGrad may be used both with recurrent neural networks and in meta-reinforcement learning, we evaluate it in a maze navigation task proposed by Miconi et al. (2018). The environment is a fixed maze and a task is defined by randomly choosing a goal location. The agent's objective is to find the location as many times as possible, being teleported to a random location each time it finds it. We use advantage actor-critic with a basic recurrent neural network (Wang et al., 2016) as the task-learner, and we design a Warp-RNN as a HyperNetwork (Ha et al., 2016) that uses an LSTM that is fixed during task training. This LSTM modulates the weights of the task-learning RNN (defined in Appendix I), which in turn is trained on mini-batches of 30 episodes for 200 000 steps. We accumulate the gradient of fixed warp-parameters continually (Algorithm 3, Appendix B) at each task parameter update. Warp parameters are updated on every 30[th]

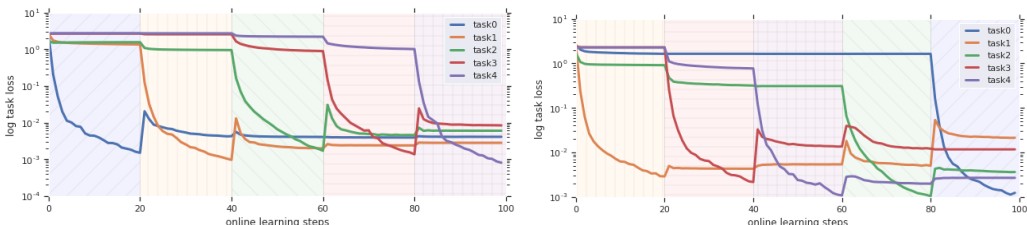

Figure 5: Continual learning experiment. Average log-loss over 100 randomly sampled tasks, each comprised of 5 sub-tasks. *Left:* learned sequentially as seen during meta-training. *Right:* learned in random order [sub-task 1, 3, 4, 2, 0].

step on task parameters (we control for meta-LSTM capacity in Appendix I). We compare against Learning to Reinforcement Learn (Wang et al., 2016) and Hebbian meta-learning (Miconi et al., 2018; 2019); see Appendix I for details. Notably, linear warps (T-Nets) do worse than the baseline RNN on this task while the Warp-RNN converges to a mean cumulative reward of ~160 in 60 000 episodes, compared to baselines that reach at most a mean cumulative reward of ~125 after 100 000 episodes (Figure 4), reaching ~150 after 200 000 episodes (I).

**(c.2) Continual Learning**   We test if WarpGrad can prevent catastrophic forgetting (French, 1999) in a continual learning scenario. To this end, we design a continual learning version of the sine regression meta-learning experiment in Finn et al. (2017) by splitting the input interval $[-5, 5] \subset \mathbb{R}$ into 5 consecutive sub-tasks (an alternative protocol was recently proposed independently by Javed & White, 2019). Each sub-task is a regression problem with the target being a mixture of two random sine waves. We train 4-layer feed-forward task-learner with interleaved warp-layers incrementally on one sub-task at a time (see Appendix J for details). To isolate the behaviour of WarpGrad parameters, we use a fixed random initialisation for each task sequence. Warp parameters are meta-learned to prevent catastrophic forgetting by defining $\mathcal{L}^\tau_{\text{meta}}$ to be the average task loss over current and previous sub-tasks, for each sub-task in a task sequence. This forces warp-parameters to disentangle the adaptation process of current and previous sub-tasks. We train on each sub-task for 20 steps, for a total of 100 task adaptation steps. We evaluate WarpGrad on 100 random tasks and find that it learns new sub-tasks well, with mean losses on an order of magnitude $10^{-3}$. When switching sub-task, performance immediately deteriorates to ~$10^{-2}$ but is stable for the remainder of training (Figure 5). Our results indicate that WarpGrad can be an effective mechanism against catastrophic forgetting, a promising avenue for further research. For detailed results, see Appendix J.

## 5   Conclusion

We propose WarpGrad, a novel meta-learner that combines the expressive capacity and flexibility of memory-based meta-learners with the inductive bias of gradient-based meta-learners. WarpGrad meta-learns to precondition gradients during task adaptation without backpropagating through the adaptation process and we find empirically that it retains the inductive bias of MAML-based few-shot learners while being able to scale to complex problems and architectures. Further, by expressing preconditioning through warp-layers that are universal function approximators, WarpGrad can express geometries beyond the block-diagonal structure of prior works.

WarpGrad provides a principled framework for general-purpose meta-learning that integrates learning paradigms, such as continual learning, an exciting avenue for future research. We introduce novel means for preconditioning, for instance with residual and recurrent warp-layers. Understanding how WarpGrad manifolds relate to second-order optimisation methods will further our understanding of gradient-based meta-learning and aid us in designing warp-layers with stronger inductive bias.

In their current form, WarpGradshare some of the limitations of many popular meta-learning approaches. While WarpGrad avoids backpropagating through the task training process, as in *Warp-Leap*, the WarpGrad objective samples from parameter trajectories and has therefore linear computational complexity in the number of adaptation steps, currently an unresolved limitation of gradient-based meta-learning. Algorithm 2 hints at exciting possibilities for overcoming this limitation.

ACKNOWLEDGEMENTS

The authors would like to thank Guillaume Desjardins for helpful discussions on an early draft as well as anonymous reviewers for their comments. SF gratefully acknowledges support from North West Doctoral Training Centre under ESRC grant ES/J500094/1 and by The Alan Turing Institute under EPSRC grant EP/N510129/1.

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

APPENDIX

## A WARPGRAD DESIGN PRINCIPLES FOR NEURAL NETS

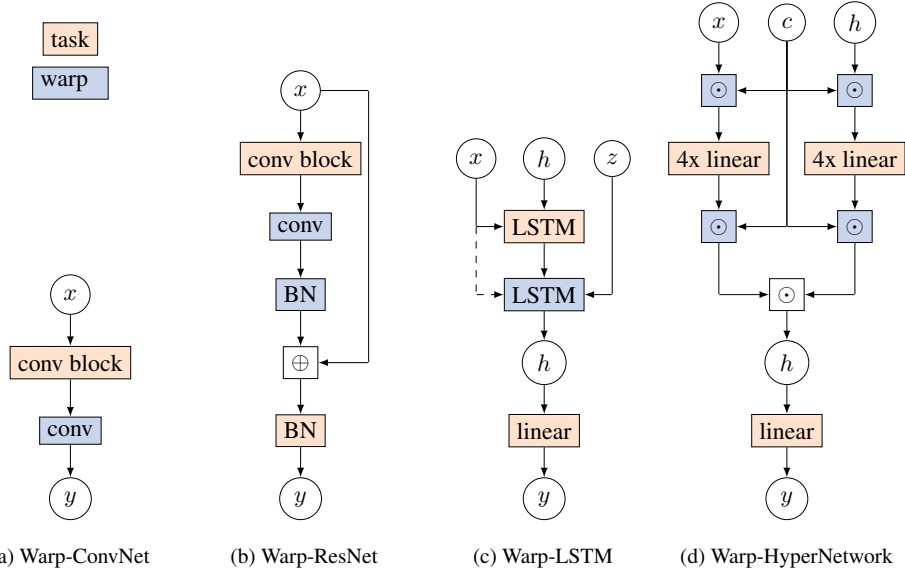

(a) Warp-ConvNet     (b) Warp-ResNet     (c) Warp-LSTM     (d) Warp-HyperNetwork

Figure 6: Illustration of possible WarpGrad architectures. Orange represents task layers and blue represents warp-layers. $\oplus$ denotes residual connections and $\odot$ any form of gating mechanism. We can obtain warped architectures by interleaving task- and warp-layers (a, c) or by designating some layers in standard architectures as task-adaptable and some as warp-layers (b, d).

WarpGrad is a model-embedded meta-learned optimiser that allows for several implementation strategies. To embed warp-layers given a task-learner architecture, we may either insert new warp-layers in the given architecture or designate some layers as warp-layers and some as task layers. We found that WarpGrad can both be used in a high-capacity mode, where task-learners are relatively weak to avoid overfitting, as well as in a low-capacity mode where task-learners are powerful and warp-layers are relatively weak. The best approach depends on the problem at hand. We highlight three approaches to designing WarpGrad optimisers, starting from a given architecture:

(a) **Model partitioning**. Given a desired architecture, designate some operations as task-adaptable and the rest as warp-layers. Task layers do not have to interleave exactly with warp-layers as gradient warping arises both through the forward pass and through backpropagation. This was how we approached the *tiered*ImageNet and *mini*ImageNet experiments.
(b) **Model augmentation**. Given a model, designate all layers as task-adaptable and interleave warp-layers. Warp-layers can be relatively weak as backpropagation through non-linear activations ensures expressive gradient warping. This was our approach to the Omniglot experiment; our main architecture interleaves linear warp-layers in a standard architecture.
(c) **Information compression**. Given a model, designate all layers as warp and interleave weak task layers. In this scenario, task-learners are prone to overfitting. Pushing capacity into the warp allows it to encode general information the task-learner can draw on during task adaptation. This approach is similar to approaches in transfer and meta-learning that restrict the number of free parameters during task training (Rebuffi et al., 2017; Lee & Choi, 2018; Zintgraf et al., 2019).

Note that in either case, once warp-layers have been chosen, standard backpropagation automatically warps gradients for us. Thus, WarpGrad is fully compatible with any architecture, for instance, Residual Neural Networks (He et al., 2016) or LSTMs. For convolutional neural networks, we may use any form of convolution, learned normalization (e.g. Ioffe & Szegedy, 2015), or adaptor module (e.g. Rebuffi et al., 2017; Perez et al., 2018) to design task and warp-layers. For recurrent networks, we can use stacked LSTMs to interleave warped layers, as well as any type of HyperNetwork

architecture (e.g. Ha et al., 2016; Suarez, 2017; Flennerhag et al., 2018) or partitioning of fast and slow weights (e.g. Mujika et al., 2017). Figure 6 illustrates this process.

## B WARPGRAD META-TRAINING ALGORITHMS

In this section, we provide the variants of WarpGrad training algorithms used in this paper. Algorithm 1 describes a simple online algorithm, which accumulates meta-gradients online during task adaptation. This algorithm has constant memory and scales linearly in the length of task trajectories. In Algorithm 2, we describe an offline meta-training algorithm. This algorithm is similar to Algorithm 1 in many respects, but differs in that we do not compute meta-gradients online during task adaptation. Instead, we accumulate them into a replay buffer of sampled task parameterisations. This buffer is a Monte-Carlo sample of the expectation in the meta objective (Eq. 13) that can be thought of as a dataset in its own right. Hence, we can apply standard mini-batching with respect to the buffer and perform mini-batch gradient descent on warp parameters. This allows us to update warp parameters several times for a given sample of task parameter trajectories, which can greatly improve data efficiency. In our Omniglot experiment, we found offline meta-training to converge faster: in fact, a mini-batch size of 1 (i.e. $\eta = 1$ in Algorithm 2 converges rapidly without any instability.

Finally, in Algorithm 3, we present a continual meta-training process where meta-training occurs throughout a stream of learning experiences. Here, $C$ represents a multi-task objective, such as the average task loss, $C^{\mathrm{multi}} = \sum_{\tau \sim p(\tau)} \mathcal{L}^{\tau}_{\mathrm{task}}$. Meta-learning arises by collecting experiences continuously (across different tasks) and using these to accumulate the meta-gradient online. Warp parameters are updated intermittently with the accumulated meta-gradient. We use this algorithm in our maze navigation experiment, where task adaptation is internalised within the RNN task-learner.

## C WARPGRAD OPTIMISERS

In this section, we detail WarpGrad methods used in our experiments.

**Warp-MAML**   We use this model for few-shot learning (Section 4.1). We use the full warp-objective in Eq. 11 together with the MAML objective (Eq. 1),

$$J^{\text{Warp-MAML}} := L(\phi) + \lambda C^{\text{MAML}}(\theta_0), \tag{14}$$

where $C^{\mathrm{MAML}} = L^{\mathrm{MAML}}$ under the constraint $P = I$. In our experiments, we trained Warp-MAML using the online training algorithm (Algorithm 1).

**Warp-Leap**   We use this model for multi-shot meta-learning. It is defined by applying Leap (Flennerhag et al., 2019) to $\theta_0$ (Eq. 16),

$$J^{\text{Warp-Leap}} := L(\phi) + \lambda C^{\text{Leap}}(\theta_0), \tag{15}$$

where the Leap objective is defined by minimising the expected cumulative chordal distance,

$$C^{\text{Leap}}(\theta_0) := \sum_{\tau \sim p(\tau)} \sum_{k=1}^{K_\tau} \left\| \mathrm{sg}\left[\vartheta^\tau_k\right] - \vartheta^\tau_{k-1} \right\|_2, \quad \vartheta^\tau_k = (\theta^\tau_{k,0}, \ldots, \theta^\tau_{k,n}, \mathcal{L}^\tau_{\mathrm{task}}(\theta^\tau_k; \phi)). \tag{16}$$

Note that the Leap meta-gradient makes a first-order approximation to avoid backpropagating through the adaptation process. It is given by

$$\nabla C^{\text{Leap}}(\theta_0) \approx - \sum_{\tau \sim p(\tau)} \sum_{k=1}^{K_\tau} \frac{\Delta \mathcal{L}^\tau_{\mathrm{task}}(\theta^\tau_k; \phi) \nabla \mathcal{L}^\tau_{\mathrm{task}}(\theta^\tau_{k-1}; \phi) + \Delta \theta^\tau_k}{\left\| \vartheta^\tau_k - \vartheta^\tau_{k-1} \right\|_2}, \tag{17}$$

where $\Delta \mathcal{L}^\tau_{\mathrm{task}}(\theta^\tau_k; \phi) := \mathcal{L}^\tau_{\mathrm{task}}(\theta^\tau_k; \phi) - \mathcal{L}^\tau_{\mathrm{task}}(\theta^\tau_{k-1}; \phi)$ and $\Delta \theta^\tau_k := \theta^\tau_k - \theta^\tau_{k-1}$. In our experiments, we train Warp-Leap using Algorithm 1 in the multi-shot *tiered*ImageNet experiment and Algorithm 2

in the Omniglot experiment. We perform an ablation study for training algorithms, comparing exact (Eq. 11) versus approximate (Eq. 12) meta-objectives, and several implementations of the warp-layers on Omniglot in Appendix F.

**Warp-RNN** For our Reinforcement Learning experiment, we define a WarpGrad optimiser by meta-learning an LSTM that modulates the weights of the task-learner (see Appendix I for details). For this algorithm, we face a continuous stream of experiences (episodes) that we meta-learn using our continual meta-training algorithm (Algorithm 3). In our experiment, both $\mathcal{L}_{\text{task}}^{\tau}$ and $\mathcal{L}_{\text{meta}}^{\tau}$ are the advantage actor-critic objective (Wang et al., 2016); $C$ is computed on one batch of 30 episodes, whereas $L$ is accumulated over $\eta = 30$ such batches, for a total of 900 episodes. As each episode involves 300 steps in the environment, we cannot apply the exact meta objective, but use the approximate meta objective (Eq. 12). Specifically, let $E^{\tau} = \{s_0, a_1, r_1, s_1, \ldots, s_T, a_T, r_T, s_{T+1}\}$ denote an episode on task $\tau$, where $s$ denotes state, $a$ action, and $r$ instantaneous reward. Denote a mini-batch of randomly sampled task episodes by $\mathbf{E} = \{E^{\tau}\}_{\tau \sim p(\tau)}$ and an ordered set of $k$ consecutive mini-batches by $\mathcal{E}^k = \{\mathbf{E}_{k-i}\}_{i=0}^{k-1}$. Then $\hat{L}(\phi; \mathcal{E}^k) = 1/n \sum_{\mathbf{E}_i \in \mathcal{E}^k} \sum_{E_{i,j}^{\tau} \in \mathbf{E}_i} \mathcal{L}_{\text{meta}}^{\tau}(\phi; \theta, E_{i,j}^{\tau})$ and

---

**Algorithm 1** WarpGrad: online meta-training

**Require:** $p(\tau)$: distribution over tasks
**Require:** $\alpha, \beta, \lambda$: hyper-parameters
1: initialise $\phi$ and $\theta_0$
2: **while** not done **do**
3:     Sample mini-batch of tasks $\mathcal{B}$ from $p(\tau)$
4:     $g_\phi, g_{\theta_0} \leftarrow 0$
5:     **for all** $\tau \in \mathcal{B}$ **do**
6:         $\theta_0^{\tau} \leftarrow \theta_0$
7:         **for all** $k$ in $0, \ldots, K_\tau - 1$ **do**
8:             $\theta_{k+1}^{\tau} \leftarrow \theta_k^{\tau} - \alpha \nabla \mathcal{L}_{\text{task}}^{\tau}(\theta_k^{\tau}; \phi)$
9:             $g_\phi \leftarrow g_\phi + \nabla L(\phi; \theta_k^{\tau})$
10:            $g_{\theta_0} \leftarrow g_{\theta_0} + \nabla C(\theta_0^{\tau}; \theta_{0:k}^{\tau})$
11:         **end for**
12:     **end for**
13:     $\phi \leftarrow \phi - \beta g_\phi$
14:     $\theta_0 \leftarrow \theta_0 - \lambda \beta g_{\theta_0}$
15: **end while**

---

**Algorithm 3** WarpGrad: continual meta-training

**Require:** $p(\tau)$: distribution over tasks
**Require:** $\alpha, \beta, \lambda, \eta$: hyper-parameters
1: initialise $\phi$ and $\theta$
2: $i, g_\phi, g_\theta \leftarrow 0$
3: **while** not done **do**
4:     Sample mini-batch of tasks $\mathcal{B}$ from $p(\tau)$
5:     **for all** $\tau \in \mathcal{B}$ **do**
6:         $g_\phi \leftarrow g_\phi + \nabla L(\phi; \theta)$
7:         $g_\theta \leftarrow g_\theta + \nabla C(\theta; \phi)$
8:     **end for**
9:     $\theta \leftarrow \theta - \lambda \beta g_\theta$
10:     $g_\theta, i \leftarrow 0, i + 1$
11:     **if** $i = \eta$ **then**
12:         $\phi \leftarrow \phi - \beta g_\phi$
13:         $i, g_\phi \leftarrow 0$
14:     **end if**
15: **end while**

---

**Algorithm 2** WarpGrad: offline meta-training

**Require:** $p(\tau)$: distribution over tasks
**Require:** $\alpha, \beta, \lambda, \eta$: hyper-parameters
1: initialise $\phi$ and $\theta_0$
2: **while** not done **do**
3:     Sample mini-batch of tasks $\mathcal{B}$ from $p(\tau)$
4:     $\mathcal{T} \leftarrow \{\tau : [\theta_0] \text{ for } \tau \text{ in } \mathcal{B}\}$
5:     **for all** $\tau \in \mathcal{B}$ **do**
6:         $\theta_0^{\tau} \leftarrow \theta_0$
7:         **for all** $k$ in $0, \ldots, K_\tau - 1$ **do**
8:             $\theta_{k+1}^{\tau} \leftarrow \theta_k^{\tau} - \alpha \nabla \mathcal{L}_{\text{task}}^{\tau}(\theta_k^{\tau}; \phi)$
9:             $\mathcal{T}[\tau].\text{append}(\theta_{k+1}^{\tau})$
10:         **end for**
11:     **end for**
12:     $i, g_\phi, g_{\theta_0} \leftarrow 0$
13:     **while** $\mathcal{T}$ not empty **do**
14:         sample $\tau, k$ without replacement
15:         $g_\phi \leftarrow g_\phi + \nabla L(\phi; \theta_k^{\tau})$
16:         $g_{\theta_0} \leftarrow g_{\theta_0} + \nabla C(\theta_0^{\tau}; \theta_{0:k}^{\tau})$
17:         $i \leftarrow i + 1$
18:         **if** $i = \eta$ **then**
19:             $\phi \leftarrow \phi - \beta g_\phi$
20:             $\theta_0 \leftarrow \theta_0 - \lambda \beta g_{\theta_0}$
21:             $i, g_\phi, g_{\theta_0} \leftarrow 0$
22:         **end if**
23:     **end while**
24: **end while**

$C^{\text{multi}}(\theta; \mathbf{E}_k) = 1/n' \sum_{E^\tau_{k,j} \in \mathbf{E}_k} \mathcal{L}^\tau_{\text{task}}(\theta; \phi, E^\tau_{k,j})$, where $n$ and $n'$ are normalising constants. The Warp-RNN objective is defined by

$$J^{\text{Warp-RNN}} := \begin{cases} L(\phi; \mathcal{E}^k) + \lambda C^{\text{multi}}(\theta; \mathbf{E}_k) & \text{if} \quad k = \eta \\ \lambda C^{\text{multi}}(\theta; \mathbf{E}_k) & \text{otherwise.} \end{cases} \tag{18}$$

**WarpGrad for Continual Learning** For this experiment, we focus on meta-learning warp-parameters. Hence, the initialisation for each task sequence is a fixed random initialisation, (i.e. $\lambda C(\theta^0) = 0$). For the warp meta-objective, we take expectations over $N$ task sequences, where each task sequence is a sequence of $T = 5$ sub-tasks that the task-learner observes one at a time; thus while the task loss is defined over the current sub-task, the meta-loss averages of the current and all prior sub-tasks, for each sub-task in the sequence. See Appendix J for detailed definitions. Importantly, because WarpGrad defines task adaptation abstractly by a probability distribution, we can readily implement a continual learning objective by modifying the joint task parameter distribution $p(\tau, \theta)$ that we use in the meta-objective (Eq. 11). A task defines a sequence of sub-tasks over which we generate parameter trajectories $\theta^\tau$. Thus, the only difference from multi-task meta-learning is that parameter trajectories are not generated under a fixed task, but arise as a function of the continual learning algorithm used for adaptation. We define the conditional distribution $p(\theta \mid \tau)$ as before by sampling sub-task parameters $\theta^{\tau_t}$ from a mini-batch of such task trajectories, keeping track of which sub-task $t$ it belongs to and which sub-tasks came before it in the given task sequence $\tau$. The meta-objective is constructed, for any sub-task parameterisation $\theta^{\tau_t}$, as $\mathcal{L}^\tau_{\text{meta}}(\theta^{\tau_t}) = 1/t \sum_{i=1}^t \mathcal{L}^\tau_{\text{task}}(\theta^{\tau_i}, \mathcal{D}_i; \phi)$, where $\mathcal{D}_j$ is data from sub-task $j$ (Appendix J). The meta-objective is an expectation over task parameterisations,

$$L^{\text{CL}}(\phi) := \sum_{\tau \sim p(\tau)} \sum_{t=1}^T \sum_{\theta^{\tau_t} \sim p(\theta|\tau_t)} \mathcal{L}^\tau_{\text{meta}}\left(\theta^{\tau_t}; \phi\right). \tag{19}$$

## D  SYNTHETIC EXPERIMENT

To build intuition for what it means to warp space, we construct a simple 2-D problem over loss surfaces. A learner is faced with the task of minimising an objective function of the form $f^\tau(x_1, x_2) = g^\tau_1(x_1) \exp(g^\tau_2(x_2)) - g^\tau_3(x_1) \exp(g^\tau_4(x_1, x_2)) - g^\tau_5 \exp(g^\tau_6(x_1))$, where each task $f^\tau$ is defined by scale and rotation functions $g^\tau$ that are randomly sampled from a predefined distribution. Specifically, each task is defined by the objective function

$$\begin{aligned} f^\tau(x_1, x_2) = {} & b^\tau_1(a^\tau_1 - x_1)^2 \exp(-x_1^2 - (x_2 + a^\tau_2)^2) \\ & - b^\tau_2(x_1/s^\tau - x_1^3 - x_2^5) \exp(-x_1^2 - x_2^2) \\ & - b^\tau_3 \exp(-(x_1 + a^\tau_3)^2 - x_1^2)), \end{aligned}$$

where each $a, b$ and $s$ are randomly sampled parameters from

$$\begin{aligned} s^\tau &\sim \text{Cat}(1, 2, \ldots, 9, 10) \\ a^\tau_i &\sim \text{Cat}(-1, 0, 1) \\ b^\tau_i &\sim \text{Cat}(-5, -4, \ldots, 4, 5). \end{aligned}$$

The task is to minimise the given objective from a randomly sampled initialisation, $x_{\{i=1,2\}} \sim U(-3, 3)$. During meta-training, we train on a task for 100 steps using a learning rate of 0.1. Each task has a unique loss-surface that the learner traverses from the randomly sampled initialisation. While each loss-surface is unique, they share an underlying structure. Thus, by meta-learning a warp over trajectories on randomly sampled loss surfaces, we expect WarpGrad to learn a warp that is close to invariant to spurious descent directions. In particular, WarpGrad should produce a smooth warped space that is quasi-convex for any given task to ensure that the task-learner finds a minimum as fast as possible regardless of initialisation.

To visualise the geometry, we use an explicit warp $\Omega$ defined by a 2-layer feed-forward network with a hidden-state size of 30 and tanh non-linearities. We train warp parameters for 100 meta-training steps;

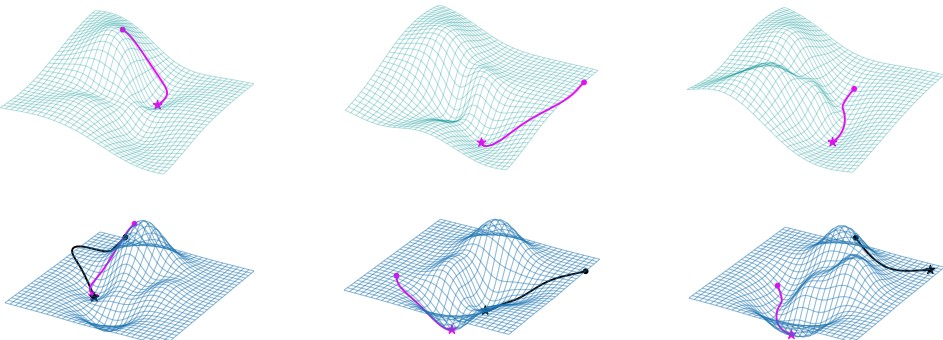

Figure 7: Example trajectories on three task loss surfaces. We start Gradient Descent (black) and WarpGrad (magenta) from the same initialisation; while SGD struggles with the curvature, the WarpGrad optimiser has learned a warp such that gradient descent in the representation space (top) leads to rapid convergence in model parameter space (bottom).

in each meta-step we sample a new task surface and a mini-batch of 10 random initialisations that we train separately. We train to convergence and accumulate the warp meta-gradient online (Algorithm 1). We evaluate against gradient descent on randomly sampled loss surfaces (Figure 7). Both optimisers start from the same initialisation, chosen such that standard gradient descent struggles; we expect the WarpGrad optimisers to learn a geometry that is robust to the initialisation (top row). This is indeed what we find; the geometry learned by WarpGrad smoothly warps the native loss surface into a well-behaved space where gradient descent converges to a local minimum.

# E OMNIGLOT

We follow the protocol of Flennerhag et al. (2019), including the choice of hyper-parameters unless otherwise noted. In this setup, each of the 50 alphabets that comprise the dataset constitutes a distinct task. Each task is treated as a 20-way classification problem. Four alphabets have fewer than 20 characters in the alphabet and are discarded, leaving us with 46 alphabets in total. 10 alphabets are held-out for final meta-testing; which alphabets are held out depend on the seed to account for variations across alphabets; we train and evaluate all baselines on 10 seeds. For each character in an alphabet, there are 20 raw samples. Of these, 5 are held out for final evaluation on the task while the remainder is used to construct a training set. Raw samples are pre-processed by random affine transformations in the form of (a) scaling between $[0.8, 1.2]$, (b) rotation $[0, 360)$, and (c) cropping height and width by a factor of $[-0.2, 0.2]$ in each dimension. This ensures tasks are too hard for few-shot learning. During task adaptation, mini-batches are sampled at random without ensuring class-balance (in contrast to few-shot classification protocols (Vinyals et al., 2016)). Note that benchmarks under this protocol are not compatible with few-shot learning benchmarks.

We use the same convolutional neural network architecture and hyper-parameters as in Flennerhag et al. (2019). This learner stacks a convolutional block comprised of a $3 \times 3$ convolution with 64 filters, followed by $2 \times 2$ max-pooling, batch-normalisation, and ReLU activation, four times. All images are down-sampled to $28 \times 28$, resulting in a $1 \times 1 \times 64$ feature map that is passed on to a final linear layer. We create a Warp Leap meta-learner that inserts warp-layers between each convolutional block, $W \circ \omega^4 \circ h^4 \circ \cdots \circ \omega^1 \circ h^1$, where each $h$ is defined as above. In our main experiment, each $\omega^i$ is simply a $3 \times 3$ convolutional layer with zero padding; in Appendix F we consider both simpler and more sophisticated versions. We find that relatively simple warp-layers do quite well. However, adding capacity does improve generalisation performance. We meta-learn the initialisation of task parameters using the Leap objective (Eq. 16), detailed in Appendix C.

Both $\mathcal{L}_{\text{meta}}^{\tau}$ and $\mathcal{L}_{\text{task}}^{\tau}$ are defined as the negative log-likelihood loss; importantly, we evaluate them on *different* batches of task data to ensure warp-layers encourage generalisation. We found no additional benefit in this experiment from using held-out data to evaluate $\mathcal{L}_{\text{meta}}^{\tau}$. We use the offline meta-training algorithm (Appendix B, Algorithm 2); in particular, during meta-training, we sample mini-batches

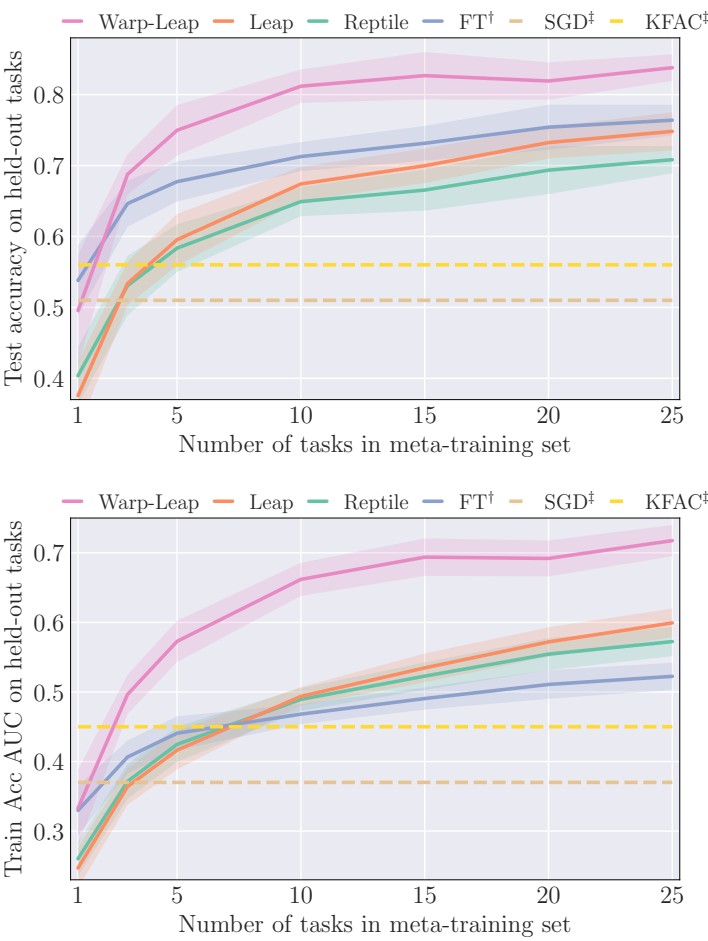

Figure 8: Omniglot results. *Top:* test accuracies on held-out tasks after meta-training on a varying number of tasks. *Bottom:* AUC under accuracy curve on held-out tasks after meta-training on a varying number of tasks. Shading represents standard deviation across 10 independent runs. We compare between Warp-Leap, Leap, and Reptile, multi-headed finetuning, as well as SGD and KFAC which used random initialisation but with a 10x larger learning rate.

of 20 tasks and train task-learners for 100 steps to collect 2000 task parameterisations into a replay buffer. Task-learners share a common initialisation and warp parameters that are held fixed during task adaptation. Once collected, we iterate over the buffer by randomly sampling mini-batches of task parameterisations without replacement. Unless otherwise noted, we used a batch size of $\eta = 1$. For each mini-batch, we update $\phi$ by applying gradient descent under the canonical meta-objective (Eq. 11), where we evaluate $\mathcal{L}_{\text{meta}}^\tau$ on a randomly sampled mini-batch of data from the corresponding task. Consequently, for each meta-batch, we take (up to) 2000 meta-gradient steps on warp parameters $\phi$. We find that this form of mini-batching causes the meta-training loop to converge much faster and induces no discernible instability.

We compare Warp-Leap against no meta-learning with standard gradient descent (SGD) or KFAC (Martens & Grosse, 2015). We also benchmark against baselines provided in Flennerhag et al. (2019); Leap, Reptile (Nichol et al., 2018), MAML, and multi-headed fine-tuning. All learners benefit substantially from large batch sizes as this enables higher learning rates. To render no-pretraining a competitive option within a fair computational budget, we allow SGD and KFAC to use 4x larger batch sizes, enabling 10x larger learning rates.

Table 2: Mean test error after 100 training steps on held out evaluation tasks. [†]Multi-headed. [‡]No meta-training, but 10x larger learning rates).

| Method No. Meta-training tasks | WarpGrad | Leap | Reptile | Finetuning[†] | MAML | KFAC[‡] | SGD[‡] |
|---|---|---|---|---|---|---|---|
| 1 | $49.5 \pm 7.8$ | $37.6 \pm 4.8$ | $40.4 \pm 4.0$ | $53.8 \pm 5.0$ | $40.0 \pm 2.6$ | **56.0** | 51.0 |
| 3 | $\mathbf{68.8} \pm 2.8$ | $53.4 \pm 3.1$ | $53.1 \pm 4.2$ | $64.6 \pm 3.3$ | $48.6 \pm 2.5$ | 56.0 | 51.0 |
| 5 | $\mathbf{75.0} \pm 3.6$ | $59.5 \pm 3.7$ | $58.3 \pm 3.3$ | $67.7 \pm 2.8$ | $51.6 \pm 3.8$ | 56.0 | 51.0 |
| 10 | $\mathbf{81.2} \pm 2.4$ | $67.4 \pm 2.4$ | $65.0 \pm 2.1$ | $71.3 \pm 2.0$ | $54.1 \pm 2.8$ | 56.0 | 51.0 |
| 15 | $\mathbf{82.7} \pm 3.3$ | $70.0 \pm 2.4$ | $66.6 \pm 2.9$ | $73.5 \pm 2.4$ | $54.8 \pm 3.4$ | 56.0 | 51.0 |
| 20 | $\mathbf{82.0} \pm 2.6$ | $73.3 \pm 2.3$ | $69.4 \pm 3.4$ | $75.4 \pm 3.2$ | $56.6 \pm 2.0$ | 56.0 | 51.0 |
| 25 | $\mathbf{83.8} \pm 1.9$ | $74.8 \pm 2.7$ | $70.8 \pm 1.9$ | $76.4 \pm 2.2$ | $56.7 \pm 2.1$ | 56.0 | 51.0 |

## F    ABLATION STUDY: WARP LAYERS, META-OBJECTIVE, AND META-TRAINING

WarpGrad provides a principled approach for model-informed meta-learning and offers several degrees of freedom. To evaluate these design choices, we conduct an ablation study on Warp-Leap where we vary the design of warp-layers as well as meta-training approaches. For the ablation study, we fixed the number of pretraining tasks to 25 and report final test accuracy over 4 independent runs. All ablations use the same hyper-parameters, except for online meta-training which uses a learning rate of 0.001.

First, we vary the meta-training protocol by (a) using the approximate objective (Eq. 12), (b) using online meta-training (Algorithm 1), and (c) whether meta-learning the learning rate used for task adaptation is beneficial in this experiment. We meta-learn a single scalar learning rate (as warp parameters can learn layer-wise scaling). Meta-gradients for the learning rate are clipped at 0.001 and we use a learning rate of 0.001. Note that when using offline meta-training, we store both task parameterisations and the momentum buffer in that phase and use them in the update rule when computing the canonical objective (Eq. 11).

Further, we vary the architecture used for warp-layers. We study simpler versions that use channel-wise scaling and more complex versions that use non-linearities and residual connections. We also evaluate a version where each warp-layer has two stacked convolutions, where the first warp convolution outputs 128 filters and the second warp convolution outputs 64 filters. Finally, in the two-layer warp-architecture, we evaluate a version that inserts a FiLM layer between the two warp convolutions. These are adapted during task training from a 0 initialisation; they amount to task embeddings that condition gradient warping on task statistics. Full results are reported in Table 3.

## G    ABLATION STUDY: WARPGRAD AND NATURAL GRADIENT DESCENT

Here, we perform ablation studies to compare the geometry that a WarpGrad optimiser learns to the geometry that Natural Gradient Descent (NGD) methods represent (approximately). For consistency, we run the ablation on Omniglot. As computing the true Fisher Information Matrix is intractable, we can compare WarpGrad against two common block-diagonal approximations, KFAC (Martens & Grosse, 2015) and Natural Neural Nets (Desjardins et al., 2015).

Table 4: Ablation study: mean test error after 100 training steps on held out evaluation tasks from a random initialisation. Mean and standard deviation over 4 seeds.

| Method | Preconditioning | Accuracy |
|---|---|---|
| SGD | None | $40.1 \pm 6.1$ |
| KFAC (NGD) | Linear (block-diagonal) | $58.2 \pm 3.2$ |
| WarpGrad | Linear (block-diagonal) | $68.0 \pm 4.4$ |
| WarpGrad | Non-linear (full) | $81.3 \pm 4.0$ |

First, we isolate the effect of warping task loss surfaces by fixing a random initialisation and only meta-learning warp parameters. That is, in this experiment, we set $\lambda C(\theta^0) = 0$. We compare against two baselines, stochastic gradient descent (SGD) and KFAC, both trained from a random initialisation. We use task mini-batch sizes of 200 and task learning rates of 1.0, otherwise we use

Table 3: Ablation study: mean test error after 100 training steps on held out evaluation tasks. Mean and standard deviation over 4 independent runs. *Offline* refers to offline meta-training (Appendix B), *online* to online meta-training Algorithm 1; *full* denotes Eq. 11 and *approx* denotes Eq. 12;[†]Batch Normalization (Ioffe & Szegedy, 2015); [‡]equivalent to FiLM layers (Perez et al., 2018);[§]Residual connection (He et al., 2016), when combined with BN, similar to the Residual Adaptor architecture (Rebuffi et al., 2017); [¶]FiLM task embeddings.

| Architecture | Meta-training | Meta-objective | Accuracy |
|---|---|---|---|
| None (Leap) | Online | None | $74.8 \pm 2.7$ |
| $3 \times 3$ conv (default) | Offline | full ($L$, Eq. 11) | $84.4 \pm 1.7$ |
| $3 \times 3$ conv | Offline | approx ($\hat{L}$, Eq. 12) | $83.1 \pm 2.7$ |
| $3 \times 3$ conv | Online | full | $76.3 \pm 2.1$ |
| $3 \times 3$ conv | Offline | full, learned $\alpha$ | $83.1 \pm 3.3$ |
| Scaling[‡] | Offline | full | $77.5 \pm 1.8$ |
| $1 \times 1$ conv | Offline | full | $79.4 \pm 2.2$ |
| $3 \times 3$ conv + ReLU | Offline | full | $83.4 \pm 1.6$ |
| $3 \times 3$ conv + BN[†] | Offline | full | $84.7 \pm 1.7$ |
| $3 \times 3$ conv + BN[†] + ReLU | Offline | full | $85.0 \pm 0.9$ |
| $3 \times 3$ conv + BN[†] + Res[§] + ReLU | Offline | full | $86.3 \pm 1.1$ |
| 2-layer $3 \times 3$ conv + BN[†] + Res[§] | Offline | full | $\mathbf{88.0} \pm 1.0$ |
| 2-layer $3 \times 3$ conv + BN[†] + Res[§] + TA[¶] | Offline | full | $\mathbf{88.1} \pm 1.0$ |

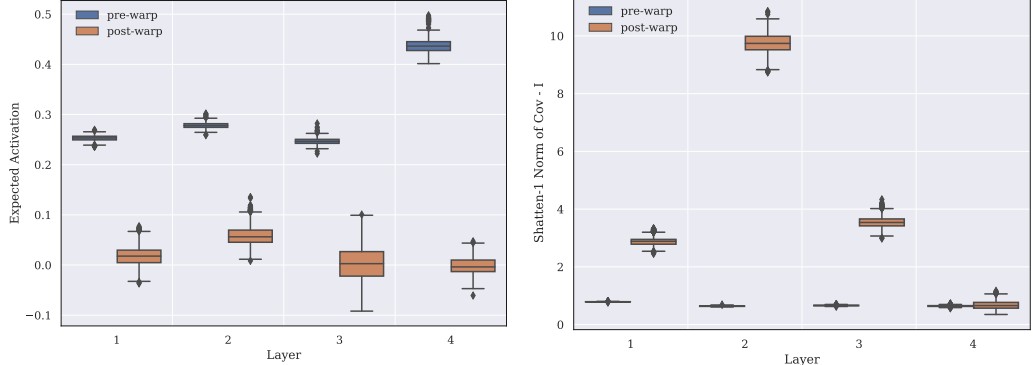

Figure 9: Ablation study. *Left:* Comparison of mean activation value $\mathbb{E}[h(x)]$ across layers, pre- and post-warping. *Right:* Shatten-1 norm of $\mathrm{Cov}(h(x), h(x)) - I$, pre- and post-norm. Statistics are gathered on held-out test set and averaged over tasks and adaptation steps.

the same hyper-parameters as in the main experiment. For WarpGrad, we meta-train with these hyper-parameters as well. We evaluate two WarpGrad architectures, in one, we use linear warp-layers, which gives a block-diagonal preconditioning, as in KFAC. In the other, we use our most expressive warp configuration from the ablation experiment in appendix F, where warp-layers are two-layer convolutional block with residual connections, batch normalisation, and ReLU activation. We find that warped geometries facilitate task adaptation on held-out tasks to a greater degree than either SGD or KFAC by a significant margin (table 4). We further find that going beyond block-diagonal preconditioning yields a significant improvement in performance.

Second, we explore whether the geometry that we meta-learn under in the full Warp-Leap algorithm is approximately Fisher. In this experiment, we use the main Warp-Leap architecture. We use a meta-learner trained on 25 tasks and that we evaluate on 10 held-out tasks. Because warp-layers are linear in this configuration, if the learned geometry is approximately Fisher, post-warp activations should be zero-centred and the layer-wise covariance matrix should satisfy $\mathrm{Cov}(\omega^{(i)}(h^{(i)}(x)), \omega^{(i)}(h^{(i)}(x))) = I$, where $I$ is the identity matrix (Desjardins et al., 2015). If true, Warp-Leap would learn a block-diagonal approximation to the Inverse Fisher Matrix, as Natural Neural Nets.

To test this, during task adaptation on held-out tasks, we compute the mean activation in each convolutional layer pre- and post-warping. We also compute the Shatten-1 norm of the difference between layer activation covariance and the identity matrix pre- and post-warping, as described above. We average statistics over task and adaptation step (we found no significant variation in these dimensions).

Figure 9 summarise our results. We find that, in general, WarpGrad-Leap has zero-centered post-warp activations. That pre-warp activations are positive is an artefact of the ReLU activation function. However, we find that the correlation structure is significantly different from what we would expect if Warp-Leap were to represent the Fisher matrix; post-warp covariances are significantly dissimilar from the identity matrix and varies across layers.

These results indicate that WarpGrad methods behave distinctly different from Natural Gradient Descent methods. One possibility is that WarpGrad methods do approximate the Fisher Information Matrix, but with higher accuracy than other methods. A more likely explanation is that WarpGrad methods encode a different geometry since they can learn to leverage global information beyond the task at hand, which enables them to express geometries that standard Natural Gradient Descent cannot.

## H  *mini*IMAGENET AND *tiered*IMAGENET

***mini*ImageNet**   This dataset is a subset of 100 classes sampled randomly from the 1000 base classes in the ILSVRC-12 training set, with 600 images for each class. Following (Ravi & Larochelle, 2016), classes are split into non-overlapping meta-training, meta-validation and meta-tests sets with 64, 16, and 20 classes in each respectively.

***tiered*ImageNet**   As described in (Ren et al., 2018), this dataset is a subset of ILSVRC-12 that stratifies 608 classes into 34 higher-level categories in the ImageNet human-curated hierarchy (Deng et al., 2009). In order to increase the separation between meta-train and meta-evaluation splits, 20 of these categories are used for meta-training, while 6 and 8 are used for meta-validation and meta-testing respectively. Slicing the class hierarchy closer to the root creates more similarity within each split, and correspondingly more diversity between splits, rendering the meta-learning problem more challenging. High-level categories are further divided into 351 classes used for meta-training, 97 for meta-validation and 160 for meta-testing, for a total of 608 base categories. All the training images in ILSVRC-12 for these base classes are used to generate problem instances for *tiered*ImageNet, of which there are a minimum of 732 and a maximum of 1300 images per class.

For all experiments, $N$-way $K$-shot classification problem instances were sampled following the standard image classification methodology for meta-learning proposed in Vinyals et al. (2016). A subset of $N$ classes was sampled at random from the corresponding split. For each class, $K$ arbitrary images were chosen without replacement to form the training dataset of that problem instance. As usual, a disjoint set of $L$ images per class were selected for the validation set.

**Few-shot classification**   In these experiments we used the established experimental protocol for evaluation in meta-validation and meta-testing: 600 task instances were selected, all using $N = 5$, $K = 1$ or $K = 5$, as specified, and $L = 15$. During meta-training we used $N = 5$, $K = 5$ or $K = 15$ respectively, and $L = 15$.

Task-learners used 4 convolutional blocks defined by with 128 filters (or less, chosen by hyper-parameter tuning), $3 \times 3$ kernels and strides set to 1, followed by batch normalisation with learned scales and offsets, a ReLU non-linearity and $2 \times 2$ max-pooling. The output of the convolutional stack ($5 \times 5 \times 128$) was flattened and mapped, using a linear layer, to the 5 output units. The last 3 convolutional layers were followed by warp-layers with 128 filters each. Only the final 3 task-layer parameters and their corresponding scale and offset batch-norm parameters were adapted during task-training, with the corresponding warp-layers and the initial convolutional layer kept fixed and meta-learned using the WarpGrad objective. Note that, with the exception of CAVIA, other baselines do worse with 128 filters as they overfit; MAML and T-Nets achieve 46% and 49 % 5-way-1-shot test accuracy with 128 filters, compared to their best reported results (48.7% and 51.7%, respectively).

Hyper-parameters were tuned independently for each condition using random grid search for highest test accuracy on meta-validation left-out tasks. Grid sizes were 50 for all experiments. We choose the optimal hyper-parameters (using early stopping at the meta-level) in terms of meta-validation test set accuracy for each condition and we report test accuracy on the meta-test set of tasks. 60000 meta-training steps were performed using meta-gradients over a single randomly selected task instances and their entire trajectories of 5 adaptation steps. Task-specific adaptation was done using stochastic gradient descent without momentum. We use Adam (Kingma & Ba, 2015) for meta-updates.

**Multi-shot classification** For these experiments we used $N = 10$, $K = 640$ and $L = 50$. Task-learners are defined similarly, but stacking 6 convolutional blocks defined by $3 \times 3$ kernels and strides set to 1, followed by batch normalisation with learned scales and offsets, a ReLU non-linearity and $2 \times 2$ max-pooling (first 5 layers). The sizes of convolutional layers were chosen by hyper-parameter tuning to $\{64, 64, 160, 160, 256, 256\}$. The output of the convolutional stack ($2 \times 2 \times 256$) was flattened and mapped, using a linear layer, to the 10 output units.

Hyper-parameters were tuned independently for each algorithm, version, and baseline using random grid search for highest test accuracy on meta-validation left-out tasks. Grid sizes were 200 for all multi-shot experiments. We choose the optimal hyper-parameters in terms of mean meta-validation test set accuracy AUC (using early stopping at the meta-level) for each condition and we report test accuracy on the meta-test set of tasks. 2000 meta-training steps were performed using averaged meta-gradients over 5 random task instances and their entire trajectories of 100 adaptation steps with batch size 64, or inner-loops. Task-specific adaptation was done using stochastic gradient descent with momentum (0.9). Meta-gradients were passed to Adam in the outer loop.

We test WarpGrad against Leap, Reptile, and training from scratch with large batches and tuned momentum. We tune all meta-learners for optimal performance on the validation set. WarpGrad outperforms all baselines both in terms of rate of convergence and final test performance (Figure 10).

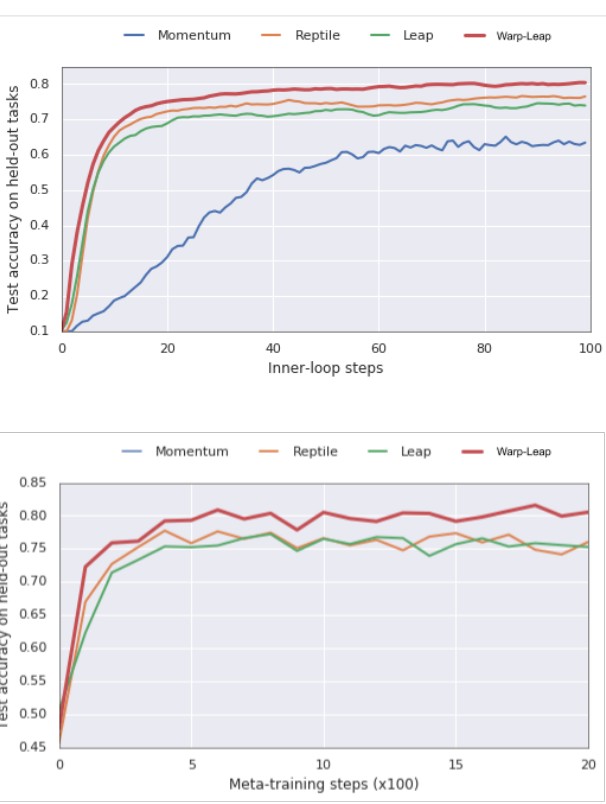

Figure 10: Multi-shot *tiered*ImageNet results. *Top:* mean learning curves (test classification accuracy) on held-out meta-test tasks. *Bottom:* mean test classification performance on held-out meta-test tasks during meta-training. Training from scratch omitted as it is not meta-trained.

## I  MAZE NAVIGATION

To illustrate both how WarpGrad may be used with Recurrent Neural Networks in an online meta-learning setting, as well as in a Reinforcement Learning environment, we evaluate it in a maze navigation task proposed by Miconi et al. (2018). The environment is a fixed maze and a task is defined by randomly choosing a goal location in the maze. During a task episode of length 200, the goal location is fixed but the agent gets teleported once it finds it. Thus, during an episode the agent must first locate the goal, then return to it as many times as possible, each time being randomly teleported to a new starting location. We use an identical setup as Miconi et al. (2019), except our grid is of size $11 \times 11$ as opposed to $9 \times 9$. We compare our Warp-RNN to Learning to Reinforcement Learn (Wang et al., 2016) and Hebbian meta-learners (Miconi et al., 2018; 2019).

The task-learner in all cases is an advantage actor-critic (Wang et al., 2016), where the actor and critic share an underlying basic RNN, whose hidden state is projected into a policy and value function by two separate linear layers. The RNN has a hidden state size of 100 and $\tanh$ non-linearities. Following (Miconi et al., 2019), for all benchmarks, we train the task-learner using Adam with a learning rate of $1e-3$ for 200 000 steps using batches of 30 episodes, each of length 200. Meta-learning arises in this setting as each episode encodes a different task, as the goal location moves, and by learning across episodes the RNN is encoding meta-information in its parameters that it can leverage during task adaptation (via its hidden state (Hochreiter & Schmidhuber, 1997; Wang et al., 2016)). See Miconi et al. (2019) for further details.

We design a Warp-RNN by introducing a warp-layer in the form of an LSTM that is frozen for most of the training process. Following Flennerhag et al. (2018), we use this meta-LSTM to modulate the task RNN. Given an episode with input vector $x_t$, the task RNN is defined by

$$h_t = \tanh \left( U_{h,t}^2 V U_{h,t}^1 \, h_{t-1} + U_{x,t}^2 W U_{x,t}^1 \, x_t + U_t^b \, b \right), \tag{20}$$

where $W$, $V$, $b$ are task-adaptable parameters; each $U_{j,t}^{(i)}$ is a diagonal warp matrix produced by projecting from the hidden state of the meta-LSTM, $U_{j,t}^{(i)} = \mathrm{diag}(\tanh(P_j^{(i)} z_t))$, where $z$ is the hidden-state of the meta-LSTM. See Flennerhag et al. (2018) for details. Thus, our Warp-RNN is a form of HyperNetwork (see Figure 6, Appendix A). Because the meta-LSTM is frozen for most of the training process, task adaptable parameters correspond to those of the baseline RNN.

To control for the capacity of the meta-LSTM, we also train a HyperRNN where the LSTM is updated with every task adaptation; we find this model does worse than the WarpGrad-RNN. We also compare the non-linear preconditioning that we obtain in our Warp-RNN to linear forms of preconditioning defined in prior works. We implement a T-Nets-RNN meta-learner, defined by embedding linear projections $T_h$, $T_x$ and $T_b$ that are meta-learned in the task RNN, $h_t = \tanh(T_h V h_t + T_x W x_t + b)$. Note that we cannot backpropagate to these meta-parameters as per the T-Nets (MAML) framework. Instead, we train $T_h, T_x, T_b$ with the meta-objective and meta-training algorithm we use for the Warp-RNN. The T-Nets-RNN does worse than the baseline RNN and generally fails to learn.

We meta-train the Warp-RNN using the continual meta-training algorithm (Algorithm 3, see Appendix B for details), which accumulates meta-gradients continuously during training. Because task training is a continuous stream of batches of episodes, we accumulating the meta-gradient using the approximate objective (Eq. 12, where $\mathcal{L}_{\mathrm{task}}^{\tau}$ and $\mathcal{L}_{\mathrm{meta}}^{\tau}$ are both the same advantage actor-critic objective) and update warp-parameters on every 30th task parameter update. We detail the meta-objective in Appendix C (see Eq. 18). Our implementation of a Warp-RNN can be seen as meta-learning "slow" weights to facilitate learning of "fast" weights (Schmidhuber, 1992; Mujika et al., 2017). Implementing Warp-RNN requires four lines of code on top of the standard training script. The task-learner is the same in all experiments with the same number of learnable parameters and hidden state size. Compared to all baselines, we find that the Warp-RNN converges faster and achieves a higher cumulative reward (Figure 4 and Figure 11).

## J  META-LEARNING FOR CONTINUAL LEARNING

Online SGD and related optimisation methods tend to adapt neural network models to the data distribution encountered last during training, usually leading to what has been termed "catastrophic

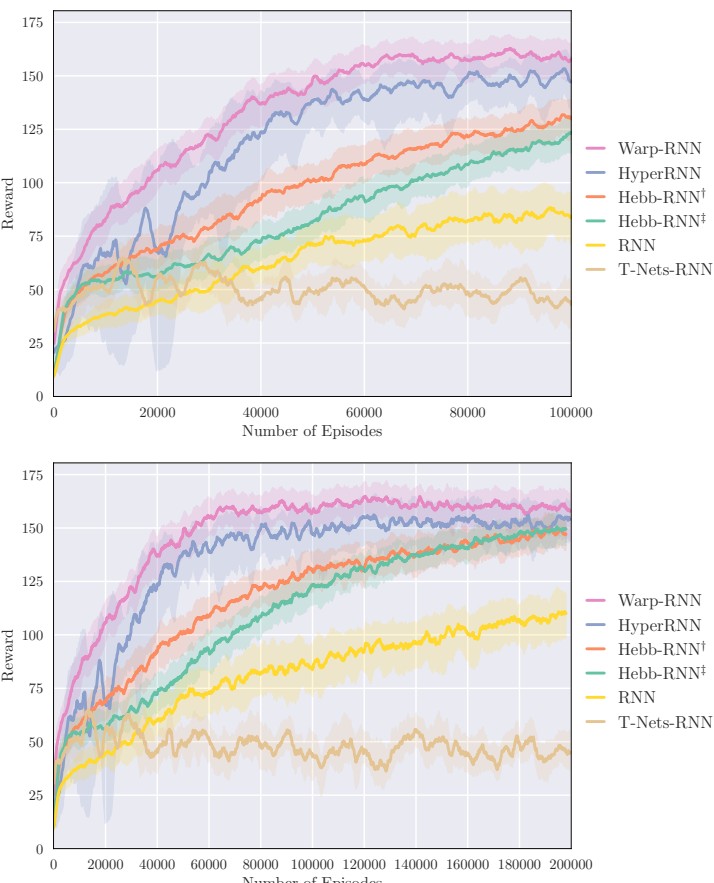

Figure 11: Mean cumulative return for maze navigation task, for 200000 training steps. Shading represents inter-quartile ranges across 10 independent runs.[†]Simple modulation and [‡]retroactive modulation, respectively (Miconi et al., 2019).

forgetting" (French, 1999). In this experiment, we investigate whether WarpGrad optimisers can meta-learn to avoid this problem altogether and directly minimise the joint objective over all tasks with every update in the fully online learning setting where no past data is retained.

**Continual Sine Regression**  We propose a continual learning version of the sine regression meta-learning experiment in Finn et al. (2017). We split the input interval $[-5, 5] \subset \mathbb{R}$ evenly into 5 consecutive sub-intervals, corresponding to 5 regression tasks. These are presented one at a time to a task-learner, which adapts to each sub-task using 20 gradient steps on data from the given sub-task only. Batch sizes were set to 5 samples. Sub-tasks thus differ in their input domain. A task sequence is defined by a target function composed of two randomly mixed sine functions of the form $f_{a_i, b_i}(x) = a_i \sin(x - b_i)$ each with randomly sampled amplitudes $a_i \in [0.1, 5]$ and phases $b_i \in [0, \pi]$. A task $\tau = (a_1, b_1, a_2, b_2, o)$ is therefore defined by sampling the parameters that specify this mixture; a task specifies a target function $g_\tau$ by

$$g_\tau(x) = \alpha_o(x)g_{a_1, b_1}(x) + (1 - \alpha_o(x))g_{a_2, b_2}(x), \tag{21}$$

where $\alpha_o(x) = \sigma(x + o)$ for a randomly sampled offset $o \in [-5, 5]$, with $\sigma$ being the sigmoid activation function.

**Model**  We define a task-learner as 4-layer feed-forward networks with hidden layer size 200 and ReLU non-linearities to learn the mapping between inputs and regression targets, $f(\cdot, \theta, \phi)$. For each task sequence $\tau$, a task-learner is initialised from a fixed random initialisation $\theta_0$ (that is not

meta-learned). Each non-linearity is followed by a residual warping block consisting of 2-layer feed-forward networks with 100 hidden units and $\mathtt{tanh}$ non-linearities, with meta-learned parameters $\phi$ which are fixed during the task adaptation process.

**Continual learning as task adaptation**  The task target function $g_\tau$ is partitioned into 5 sets of sub-tasks. The task-learner sees one partition at a time and is given $n = 20$ gradient steps to adapt, for a total of $K = 100$ steps of online gradient descent updates for the full task sequence; recall that every such sequence starts from a fixed random initialisation $\theta_0$. The adaptation is completely online since at step $k = 1, \dots, K$ we sample a new mini-batch $D_{\text{task}}^k$ of 5 samples from a single sub-task (sub-interval). The data distribution changes after each $n = 20$ steps with inputs $x$ coming from the next sub-interval and targets form the same function $g_\tau(x)$. During meta-training we always present tasks in the same order, presenting intervals from left to right. The online (sub-)task loss is defined on the current mini-batch $D_{\text{task}}^k$ at step $k$:

$$\mathcal{L}_{\text{task}}^\tau \left( \theta_k^\tau, D_{\text{task}}^k; \phi \right) = \frac{1}{2|\mathcal{D}_{\text{task}}^k|} \sum_{x \in D_{\text{task}}^k} \left( f(x, \theta_k^\tau; \phi) - g_\tau(x) \right)^2. \tag{22}$$

Adaptation to each sub-task uses sub-task data only to form task parameter updates $\theta_{k+1}^\tau \leftarrow \theta_k^\tau - \alpha \nabla \mathcal{L}_{\text{task}}^\tau \left( \theta_k^\tau, D_{\text{task}}^k; \phi \right)$. We used a constant learning rate $\alpha = 0.001$. Warp-parameters $\phi$ are fixed across the full task sequence during adaptation and are meta-learned across random samples of task sequences, which we describe next.

**Meta-learning an optimiser for continual learning**  To investigate the ability of WarpGrad to learn an optimiser for continual learning that mitigates catastrophic forgetting, we fix a random initialisation prior to meta-training that is not meta-learned; every task-learner is initialised with these parameters. To meta-learn an optimiser for continual learning, we need a meta-objective that encourages such behaviour. Here, we take a first step towards a framework for meta-learned continual learning. We define the meta-objective $\mathcal{L}_{\text{meta}}^\tau$ as an incremental multitask objective that, for each sub-task! $\tau_t$ in a given task sequence $\tau$, averages the validation sub-task losses (Eq. 22) for the current and every preceding loss in the task sequence. The task meta-objective is defined by summing over all sub-tasks in the task sequence. For some sub-task parameterisation $\theta^{\tau_t}$, we have

$$\mathcal{L}_{\text{meta}}^\tau \left( \theta^{\tau_t}; \phi \right) = \sum_{i=1}^t \frac{1}{n(T - t + 1)} \mathcal{L}_{\text{task}}^\tau \left( \theta^{\tau_i}, D_{\text{val}}^i; \phi \right). \tag{23}$$

As before, the full meta-objective is an expectation over the joint task parameter distribution (Eq. 11); for further details on the meta-objective, see Appendix C, Eq. 19. This meta-objective gives equal weight to all the tasks in the sequence by averaging the regression step loss over all sub-tasks where a prior sub-task should be learned or remembered. For example, losses from the first sub-task, defined using the interval $[-5, -3]$, will appear $nT$ times in the meta-objective. Conversely, the last sub-task in a sequence, defined on the interval $[3, 5]$, is learned only in the last $n = 20$ steps of task adaptation, and hence appears $n$ times in the meta-objective. Normalising on number of appearances corrects for this bias. We trained warp-parameters using Adam and a meta-learning rate of $0.001$, sampling 5 random tasks to form a meta-batch and repeating the process for $20\,000$ steps of meta-training.

**Results**  Figure 12 shows a breakdown of the validation loss across the 5 sequentially learned tasks over the 100 steps of online learning during task adaptation. Results are averaged over 100 random regression problem instances. The meta-learned WarpGrad optimiser reduces the loss of the task currently being learned in each interval while also largely retaining performance on previous tasks. There is an immediate relatively minor loss of performance, after which performance on previous tasks is retained. We hypothesise that this is because the meta-objectives averages over the full learning curve, as opposed to only the performance once a task has been adapted to. As such, the WarpGrad optimiser may allow for some degree of performance loss. Intriguingly, in all cases, after an initial spike in previous sub-task losses when switching to a new task, the spike starts to revert back some way towards optimal performance, suggesting that the WarpGrad optimiser facilitates positive backward transfer, without this being explicitly enforced in the meta-objective. Deriving a principled meta-objective for continual learning is an exciting area for future research.

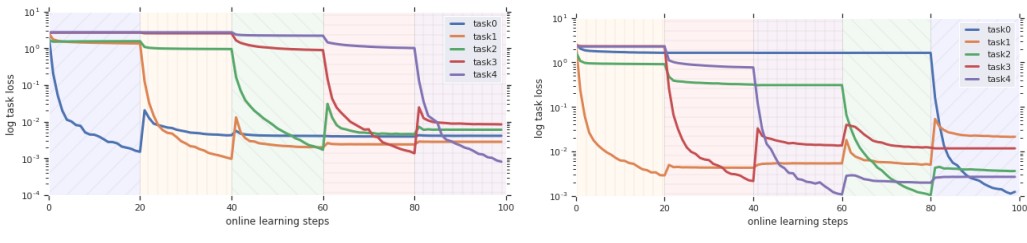

(a) Task order as seen during meta-training.                    (b) Random task order.

Figure 12: Continual learning regression experiment. Average log-loss over 100 randomly sampled tasks. Each task contains 5 sub-tasks learned (a) sequentially as seen during meta-training or (b) in random order [sub-task 1, 3, 4, 2, 0]. We train on each sub-task for 20 steps, for a total of $K = 100$ task adaptation steps.

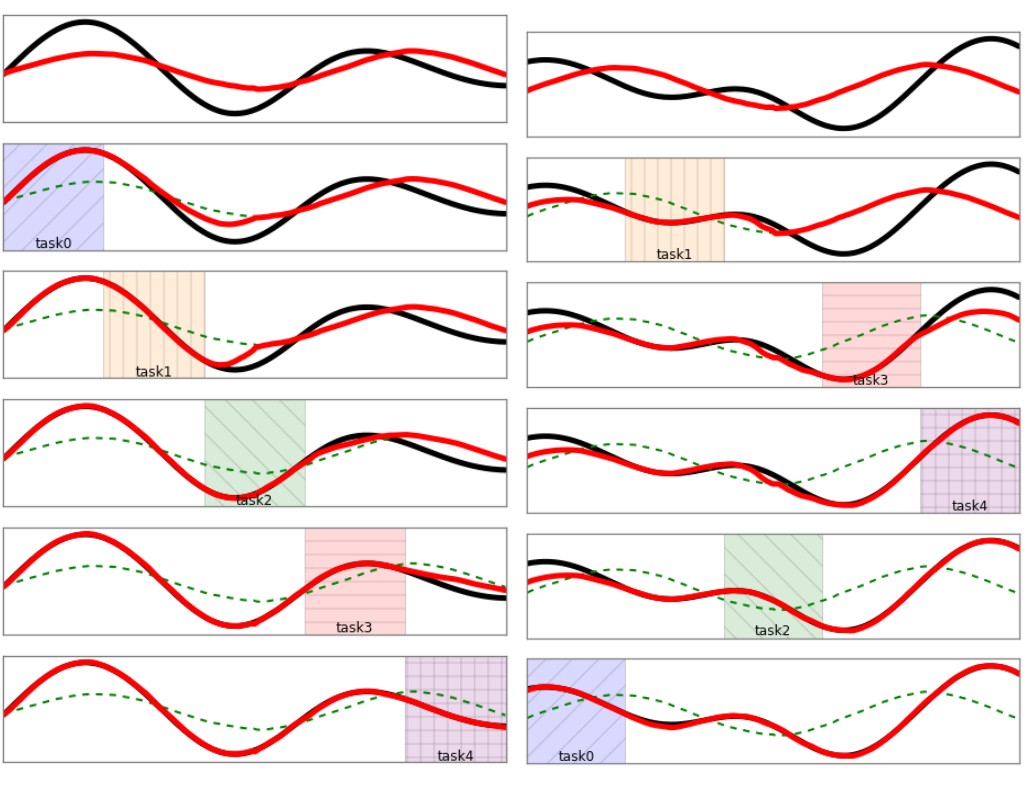

(a) Task order seen during meta-training.                    (b) Random task order.

Figure 13: Continual learning regression: evaluation after partial task adaptation. We plot the ground truth (black), task-learner prediction before adaptation (dashed green) and task-learner prediction after adaptation (red). Each row illustrates how task-learner predictions evolve (red) after training on sub-tasks up to and including that sub-task (current task illustrate in plot). (a) sub-tasks are presented in the same order as seen during meta-training; (b) sub-tasks are presented in random order at meta-test time in sub-task order [1, 3, 4, 2 and 0].

