# OpenReview forum: "Meta-Learning with Warped Gradient Descent"
_ICLR.cc/2020/Conference — Accept (Talk)_

### Official Review · AnonReviewer1 · 2019-10-23
**Official Blind Review #1**

**Rating:** 8

**Review:**

This paper proposes a learning strategy to precondition gradients for meta-learning. I really enjoyed reading the paper though I admit that I couldn't fully grasp all the details yet (paper is dense). My comments below are mostly to improve the readability of the paper for readers like me (knowing a thing or two in optimization and meta-learning)



1- The authors emphasize on the method being trajectory-agnostic. Can you explain why this is very important? What methods are not trajectory-agnostic?

2 - Also in various places, the authors claim the method does not suffer from vanishing/exploding gradients and credit-assignment problem. This needs to be properly verified (and explained as I do not see the connections clearly)

3- Some claims are based on the Omniglot experiments (eg., the effect of the stop-gradient). It would be good if this can be done on Mini-imagenet instead.

4- I am not sure I understand the stop-gradient operator, can you be more explicit there?

5- I read the conversation regarding linear units on openreview and I disagree with your statement. A cascade of linear layers does not necessarily match one linear layer unless some constraints on the rank of layers are envisaged, a bottleneck in the middle ruin everything.


**Experience Assessment:**

I have read many papers in this area.

**Review Assessment: Checking Correctness Of Derivations And Theory:**

I assessed the sensibility of the derivations and theory.

**Review Assessment: Checking Correctness Of Experiments:**

I assessed the sensibility of the experiments.

**Review Assessment: Thoroughness In Paper Reading:**

I read the paper at least twice and used my best judgement in assessing the paper.

---

> ### Author Response · Authors · 2019-11-08
> **Response to Review #1**
>
> Dear R1,
>
> Thank you for your thorough review and constructive feedback, we will incorporate these when updating the manuscript to make it as accessible as possible!
>
> 1 - This is a great point, it is indeed important and we will make sure to emphasise this in the revised manuscript. A meta-learner that is *not* trajectory agnostic has a meta-objective that is a function of the entire trajectory, and hence needs to backpropagate through the entire trajectory, such as MAML-based meta-learners. This limits their scalability and makes meta-optimization challenging (see Eq. 1 and following discussion). In contrast, WarpGrad is a trajectory-agnostic meta-learner. We use trajectories to form an empirical distribution from which we sample individual steps that we optimise independently. Because the objective is point-wise, we avoid backpropagation through the trajectory, which is what makes WarpGrad competitive even for very long trajectories: the meta-objective scales with at most linear complexity in trajectory length and does not suffer numerical instability as trajectories become long.
>
> 2 - Thank you for the constructive feedback as this too is a central aspect of WarpGrad. To clarify the connection, MAML backpropages through the adaptation trajectory, which is essentially an RNN-like backpropagation through time operation. Hence it suffers from the same exploding/vanishing gradients and credit assignment problems that RNNs struggle with, which has been observed empirically [e.g. 1]. More specifically, because the MAML meta-gradient is a product of Hessian matrices, high or low curvature during task adaptation has a multiplicative effect on the meta-gradient (the product of many values much greater or much lower than 1 will cause gradients to explode or vanish, respectively). In contrast, WarpGrad avoids these specific issues by design as it does not backpropagate through adaptation trajectories and instead learns to optimize each gradient step individually.
>
> 3 - While we appreciate the sentiment, tieredImageNet, miniImagenet and Omniglot are structurally similar benchmarks in that they are image classification tasks over homogenous image domains (natural images and hand-drawn characters, respectively). Hence the added benefit of running the same ablations on miniImagenet would be limited and given space as well as time constraints we have refrained from doing so. In terms of the first-order approximation (Eq. 12), our claim is that it is a useful approximation over longer adaptation processes, as in the Omniglot and RL experiment, where first-order effects tend to dominate. Hence we evaluate this approximation on these experiments.
>
> 4 - A stop-gradient operator is an operation that prevents gradients from flowing through a variable during backpropagation. We use it in Eq. 12 to make the same approximation as in the first-order approximation of MAML [2], where the stop-gradient operator prevents the meta-gradient from backpropagating through the inner task adaptation step. That way, the meta-gradient avoids computing second-order derivatives (that is, it renders the meta-gradient Hessian-free).
>
> 5 - As R1 correctly points out, we are making an implicit full-rank assumption. The public comment was concerned with potentially unfair comparisons in the case that warp layers increase model capacity. Our reply was directed towards this concern. While linear layers cannot add capacity, they can reduce it. This would not make comparisons to baselines unfair, though potentially unfavorable (hence our results are erring on the side of caution) for WarpGrad. Note that all baselines are tuned for model capacity through conv-layer filter sizes.
>
> [1] Finn et. al. Model-Agnostic Meta-Learning for Fast Adaptation of Deep Networks. 2017.
> [2] Antoniou et. al. How to train your MAML. 2019.

---

### Official Review · AnonReviewer3 · 2019-10-23
**Official Blind Review #3**

**Rating:** 8

**Review:**

The authors propose warped gradient descent (WarpGrad) an optimisation framework for facilitating gradient-based meta-learning. WarpGrad interleaves within the learner meta-learned warp-layers that implicitly precondition the gradients of the task-specific parameters during backpropagation. In contrast to the linear projection layers employed in T-Nets, warp-layers are unrestricted in form and induce a full Jacobian preconditioning matrix. The warp layers are meta-learned in a trajectory-agnostic fashion, thus obviating the need to backpropagate through the gradient steps to compute the updates of their parameters. The framework is readily applicable to standard gradient-based meta-learners, and is shown to yield a significant boost in performance on both few-shot and multi-shot learning tasks, as well as to have promising applications to continual learning.

The paper is well-structured and well-motivated: the problem statement is clearly laid out from the outset, with appropriate context, and explanations supported well diagramatically. The idea, and perhaps more so the applications thereof, is seemingly novel and its explanation is given straightforwardly while avoiding getting bogged down in technical details. Clear comparisons and distinctions with previous work are drawn - for instance with the update rules for several gradient-based methods - MAML and its derivatives - being laid out in standard form (though it might also be nice to echo this with the WarpGrad update rule).

The experiments are logically ordered with the initial set covering the standard few-shot learning benchmarks with appropriate baselines (though the results for few-shot tieredImageNet are lacking in this respect), with most essential details given in the main text and full details, including those related to the datasets in question and hyperparameter selection, documented in Appendix H. Meta-learning does seem uniquely well-positioned for tackling the task of continual learning and it's heartening to see this being explored here with a degree of success - it would be interested to see how its performance compares with standard continual learning methods (such as EWC) on the same task. Particularly impressive is the depth into which the Appendices regarding the experiments, both elaborating on the details given in the main text as well as additional ablation studies.

Minor errors:

- Page 7: "a neural network that dynamically **adapt** the parameters..." - should be "adapts"
- Page 22: "where $I$ is the **identify** matrix" - should be "identity"
- Page 27: "The task target function $g_\tau$ is **partition** into 5 sets of **sub-task**" - should be "partitioned" and "sub-tasks", respectively

**Experience Assessment:**

I have read many papers in this area.

**Review Assessment: Checking Correctness Of Derivations And Theory:**

I carefully checked the derivations and theory.

**Review Assessment: Checking Correctness Of Experiments:**

I carefully checked the experiments.

**Review Assessment: Thoroughness In Paper Reading:**

I read the paper at least twice and used my best judgement in assessing the paper.

---

> ### Author Response · Authors · 2019-11-08
> **Response to Review #3**
>
> Dear R3,
>
> Thank you for your detailed review and thoughtful comments! We will incorporate them when revising the manuscript. We agree that meta-learning to learn continually is an exciting new area of research and are thrilled to report a positive signal. Due to space constraints, we will not be able to delve deeper in this paper, but are certainly excited to push further in this direction!

---

### Official Review · AnonReviewer2 · 2019-11-03
**Official Blind Review #2**

**Rating:** 8

**Review:**

Summary:
The current paper deals with meta-learning and essentially proposes a generalization of MAML (a popular gradient-based meta-learning algorithm) that mostly builds upon two main recent advances in meta-learning: 1) an architectural one (see e.g. T-Nets), which consists in optimizing the parameters of additional layers during the meta-learning outer loop (as opposed to only optimizing the initial conditions of the original parameters like in MAML), and 2) a theoretical one (see e.g. Meta-SGD, Meta-curvature), which is based on the geometrical observation that one set of parameters can precondition a second set of parameters that are consequently being optimized in a "warped" geometry, possibly speeding up learning.
The authors provide a great and thorough overview of the literature, in particular for gradient-based meta-learning methods, which helps putting all this in perspective.
The way they obtain the mentioned "warped" geometry in practice is by adding additional so-called warp-layers to an architecture that is being trained with meta-learning. Such warp-layer are generic deep learning modules (such as convolutions followed by BatchNorm, or LSTM layers), which are being trained in the outer-loop of the meta-learning optimization. In this sense, WarpGrad extend T-Nets, which only allowed for linear layers.
The second main innovation of WarpGrad is the proposal of a new meta-learning objective, which incorporates a meta-learning internal loop of only one step of (preconditioned) SGD, meaning that, as the authors notes, "in contrast to MAML-based approaches (Eq. 1), [...] avoids backpropagation through learning processes".
The authors test their algorithm on several meta-learning benchmarks, including few- and multi-shot learning tasks demonstrating very competitive performance when their algorithm is combined with MAML or Leap. They then deploy WarpGrad on a maze navigation reinforcemente learning task to demonstrate training of recurrent architectures, and on a continual learning toy dataset to show that their objective can be adapted to mitigate catastrophic forgetting.

Decision:
This is a good paper which proposes an interesting generalization of previous gradient-based meta-learning methods like MAML and T-Net, with an impressive number of experiments. However, some of the statements regarding the advantages of WarpGrad over previous algorithms seem a little bit misleading, in particular in situations where WarpGrad needs to be combined with these same algorithms. For instance (and I might have completely misunderstood things here), it seems that when the WarpGrad objective is being combined with MAML (which requires backpropagation through multiple-step gradient descent trajectories), then also the resulting combined objective will necessarily need to backprop through the same multi-step trajectory, defeating the stated advantage of the WarpGrad algorithm (i.e. that its objective avoids backpropagating through the learning processes).
In general, even if one only considers the WarpGrad objective eq. (10), that comprises a meta-learning inner loop which consists of one step of (preconditioned) gradient descent. However, it seems like an arbitrary (and limiting) choice of the authors to only perform one step, as opposed to multiple ones. As a matter of fact, even very sophisticated second order gradient descent methods like natural gradient descent typically require more than one step to reach a local minimum. That is to say, that the main advantage showcased by the authors (the fact that the WarpGrad objective avoids backprop through a whole learning trajectory) seems like a limitation, rather than the result of a principled derivation.
It would be beneficial if the authors could clarify this points. In particular, whether combining WarpGrad with MAML does not indeed negate the stated advantages of WarpGrad over MAML, and whether there is a principled way of demonstrating that executing only one step in the inner loop of the WarpGrad objective is completely general (i.e., additional steps do not help the inner loop).

Minor:
- The authors use the wrong citation key when referring to the T-net paper: it should be Lee et al 2018, instead of Lee et al. 2017
- I believe that when the authors mention Fast and slow weights, they are being described in the opposite way: slow weights should be in charge of meta-learning information, while fast ones are in charge of task-specific information.
- Line 3 and 4 of Algorithm 1 and 2: shouldn't it say "mini-batch of tasks" (plural), instead of "mini-batch of task", since several tasks are being sampled? Otherwise, it might be erroneously interpreted as "mini-batch of (samples belonging to) task T".
- The comment that "learning to precondition gradients can be seen as a Markov Process of order 1" is never clearly elucidated or developed. It would help to develop this.

**Experience Assessment:**

I have published one or two papers in this area.

**Review Assessment: Checking Correctness Of Derivations And Theory:**

I assessed the sensibility of the derivations and theory.

**Review Assessment: Checking Correctness Of Experiments:**

I assessed the sensibility of the experiments.

**Review Assessment: Thoroughness In Paper Reading:**

I read the paper thoroughly.

---

> ### Author Response · Authors · 2019-11-08
> **Response to Review #2**
>
> Dear R2,
>
> Thank you for your thoughtful review and comprehensive feedback! We will revise the manuscript to address all of your concerns, as detailed below.
>
> [WarpGrad extend T-Nets, which only allowed for linear layers....]
>
> WarpGrad does indeed extend the T-Nets architecture in this way, we would like to emphasise that this extension is motivated by a subtle but important theoretical aspect of non-linearity in warp layers. Theoretically speaking, the meta-objective relies on non-linear gradient preconditioning. As we are taking a trajectory agnostic approach, the meta-learner should be able to modulate preconditioning on task data to ensure taking an expectation over parameter space generate useful preconditioning across tasks and adaptation steps. This requires non-linear warp layers, and thus the meta-objective and the architectural contribution are tied on a deeper theoretical level.
>
> In practice, linear warp layers do work quite well for supervised learning, but on the other hand, as we show in the ablation study in Appendix G, if we make warp-layers non-linear, we get similar performance from a *random* initialisation. For more complex tasks, as in the RL case, we show that non-linear warp layers are crucial (detailed in Appendix I).
>
> [In general, even if one only considers the WarpGrad objective...]
>
> We believe there may be some misunderstanding here, as WarpGrad does not limit the inner loop to one gradient step; in fact, it is independent of the number of inner steps. The canonical WarpGrad objective (Eq. 10) is the expected one-step gradient update over a joint distribution of objective functions (L) and model parameters (\gamma): it is a global objective defined in terms of the vector field of the manifold W. Put simply, Eq. 10 solves for good preconditioning over all of parameter space, irrespective of how many steps of adaptation we are taking on some objective L. In practice, we approximate the distribution in Eq. 10 - as we detail in the paragraph between Eq. 10 and Eq. 11 - by constructing a Monte-Carlo estimator on the trajectories collected over  K-steps of adaptation in the inner loop (see also Algorithm 1 and 2). We optimise individual steps sampled from the estimator, which effectively allows WarpGrad to be unaffected by the inner step size. Similarly, at meta-test time, WarpGrad is compatible with any number of adaptation steps. In our experiments, we use the same K for meta-training and testing: for instance, on the Omniglot experiment, the inner loop during meta-training and meta-testing use 100 steps of task adaptation. The WarpGrad objective (Eq. 11) is an expectation over these one-step parameter updates (see also Algorithm 1 and 2). We hope this clarifies, if we misunderstood this concern please do let us know.
>
> [statements regarding the advantages of WarpGrad...]
>
> We appreciate this comment and sympathise with the reviewer’s concern. We will clarify in the revised manuscript that WarpGrad does *not* “need to be combined with” some learned initialisation like MAML or Leap—we do so to identify the effect of the WarpGrad objective on miniImagenet and Omniglot, respectively. As R2 points out in the summary, we make two contributions: one architectural and one algorithmic. We combine WarpGrad with MAML and Leap to obtain all-else-equals comparisons of the meta-objective. We show in the Omniglot ablation study (Appendix F) that non-linear warp-layers can perform on par even from random initialisations. In the RL experiment, neither MAML nor Leap are well-defined, but applying WarpGrad is straightforward.
>
> [WarpGrad objective combined with MAML...]
>
> As R2 correctly points out, when combined with MAML, the scalability advantage of WarpGrad is lost, but we retain its geometrical properties as well as other numerical properties (e.g. stability) with respect to warp layers. Hence while Warp-MAML does not enjoy the scalability advantage, it does retain all other properties of WarpGrad. MAML is a powerful algorithm for few-shot learning problems where we can afford to backpropagate through the adaptation process and we find that the combination of WarpGrad and MAML compares favorably to pure MAML-based preconditioning. For other meta-learning problems that are not few-shot, we show that WarpGrad can be used effectively without backpropagation through the adaptation process on a variety of large-scale meta-learning benchmarks.
>
> We hope that our replies resolve above concerns and we will update our manuscript to emphasise that we use few/multi-shot learning to evaluate the meta-objective, holding the architecture and initialisation fixed. We measure the effect of our meta-architecture through ablations both with and without a meta-learned initialisation and demonstrate their combined effectiveness on complex meta-learning tasks in RL and continual learning.

---

### Public Comment · ~Zhijie_Deng1 · 2019-10-11
**Question about preconditioning**

Nice work! I have a short question. If I understand correctly, the gradient modifier w should only stay in the back-propagation path, but it also stays in the forward-propagation path in your work. Can it be removed and why?

---

> ### Author Response · Authors · 2019-10-11
> **Re: Question about preconditioning**
>
> Thank you, and great that you reached out!
>
> When preconditioning is defined by explicitly projecting a parameter gradient via some (smoothly varying) matrix [e.g. 1, 2, 3], that matrix would not be part of the forward pass of the model.
>
> However, we can also think of preconditioning as inserting layers w that ‘warp’ model parameters in the forward pass - backpropagating through such warp layers automatically preconditions the gradient [e.g. 4, 5, our work]. In this case, we do want w to be part of the forward pass.
>
> Both perspectives describe preconditioning but in different ways. In this work, we argue that warp layers is a more effective approach because it interacts with the model both in the forward and backward pass while being simple to implement.
>
> Hope that answers your question!
>
> =========
>
> References
>
> 1. Amari. Natural gradient works efficiently in learning. Neural computation 10.2. 251-276. 1998.
> 2. Li et. al.. Meta-SGD: Learning to Learn Quickly for Few-Shot Learning. ArXiv 1707.09835. 2017.
> 3. Park et. al.. Meta-Curvature. NeurIPS. 2019.
> 4. Desjardins et. al.. Natural Neural Networks.Neurips. 2016.
> 5. Lee et. al.. Gradient-Based Meta-Learning with Learned Layerwise Metric and Subspace. ICML. 2017.

---

> > ### Public Comment · ~Zhijie_Deng1 · 2019-10-12
> > **Re: Re: Question about preconditioning**
> >
> > Thanks for the instant reply. A follow-up question: when using the preconditioning layers in the forward pass, the task learner essentially adopts a deeper network for specific tasks, compared to the standard methods (e.g., MAML, MC). Although the additional layers are trained with meta objective instead of the task objective, they may enhance the expressive ability of the task learner, so are the performance comparisons in the experiment section (e.g., Table 1) unfair?

---

> > > ### Author Response · Authors · 2019-10-14
> > > **Re: Question about preconditioning**
> > >
> > > Thank you for your question and valid concern. In general, interleaving nonlinear warp layers does increase the depth of the task learner. However, when warp layers are linear, the effective depth of the task learner does not increase since the composition of two linear transformations is a linear transformation itself. In other words, when warp layers are linear, they can be seen as part of existing layers in the task learner (akin to [4, 5] above).
> > >
> > > To the extent possible, experiments have been carefully designed to ensure fair comparisons. All results reported in Table 1 use linear warp layers and baselines have been hyper-parameter tuned with equal computational budgets.
> > >
> > > With that said, it is worth noting that non-linear warp-layers improve performance considerably. We report such results inline in the main text as well as in detailed ablation studies (see Appendix F, G, and J). We hope this addresses your concern.

---

### Decision · Program_Chairs · 2019-12-19

**Decision:**

Accept (Talk)

**Comment:**

A strong paper reporting improved approaches to meta-learning.